# Transcriptional profiling of mouse peripheral nerves to the single-cell level to build a sciatic nerve ATlas (SNAT)

**Daniel Gerber[1], Jorge A Pereira[1], Joanne Gerber[1], Ge Tan[2], Slavica Dimitrieva[2], Emilio Yánguez[2], Ueli Suter[1]***

[1]Department of Biology, Institute of Molecular Health Sciences, Swiss Federal Institute of Technology, ETH Zurich, Zurich, Switzerland; [2]Functional Genomics Center Zurich, ETH Zurich/University of Zurich, Zurich, Switzerland

**Abstract** Peripheral nerves are organ-like structures containing diverse cell types to optimize function. This interactive assembly includes mostly axon-associated Schwann cells, but also endothelial cells of supporting blood vessels, immune system-associated cells, barrier-forming cells of the perineurium surrounding and protecting nerve fascicles, and connective tissue-resident cells within the intra-fascicular endoneurium and inter-fascicular epineurium. We have established transcriptional profiles of mouse sciatic nerve-inhabitant cells to foster the fundamental understanding of peripheral nerves. To achieve this goal, we have combined bulk RNA sequencing of developing sciatic nerves up to the adult with focused bulk and single-cell RNA sequencing of Schwann cells throughout postnatal development, extended by single-cell transcriptome analysis of the full sciatic nerve both perinatally and in the adult. The results were merged in the transcriptome resource Sciatic Nerve ATlas (SNAT: https://www.snat.ethz.ch). We anticipate that insights gained from our multi-layered analysis will serve as valuable interactive reference point to guide future studies.

**\*For correspondence:**
usuter@cell.biol.ethz.ch

**Competing interests:** The authors declare that no competing interests exist.

## Introduction

Peripheral nerves are specialized in the transport of information between the central nervous system and peripheral body parts, including muscles and sensory cells (*Stassart et al., 2018*). This function is mainly fulfilled by axons/neurons in symbiosis with interacting Schwann cells (SCs). In addition, other cell types contribute crucially to ensure optimized nerve structure, function, and support. This group includes cells expressing *Pdgfra* (encoding Platelet Derived Growth Factor Receptor Alpha) such as endoneurial, perineurial, and epineural cells (also designated as fibroblast-like or mesenchymal cells [*Carr et al., 2019*; *Joseph et al., 2004*; *Ma et al., 2018*]), vascular cells (such as endothelial cells, pericytes, and vascular smooth muscle cells) (*Carr et al., 2019*; *Stierli et al., 2018*; *Tserentsoodol et al., 1999*), cells of the immune system (*Carr et al., 2019*; *Hidmark et al., 2017*; *Stierli et al., 2018*), and epineurial adipocytes (*Montani et al., 2018*; *Nadra et al., 2012*).

Each of the nerve-resident cell types is expected to express a distinct portfolio of genes at particular levels, depending on the differentiation state during development, degeneration and regeneration after injury, and in aging. However, our knowledge in this context is fragmentary and often limited in cellular resolution. For example, several transcriptome analyses in mice and rats have been carried out that reflect peripheral nerves as an organ (*Arthur-Farraj et al., 2017*; *D'Antonio et al., 2006*; *Figlia et al., 2017*; *Fledrich et al., 2018*; *Fröb et al., 2019*; *Gerber et al., 2019*; *Gökbuget et al., 2018*; *Kim et al., 2018*; *Le et al., 2005*; *Nagarajan et al., 2002*; *Patzig et al., 2011*; *Quintes et al., 2016*; *Ryu et al., 2008*; *Verheijen et al., 2003*), complemented by some proteomics studies (*Patzig et al., 2011*; *Siems et al., 2020*). Albeit valuable, these studies cannot

distinguish between particularities of the different cell populations of the different tissues involved. They are also prone to some misinterpretations due to the limitations of bulk analysis of cellularly heterogenous organs and tissues. Nevertheless, since SCs are the most numerous cell type present in nerves (*Stierli et al., 2018*), such datasets can be and have been interpreted, at first approximation, as a surrogate for the analysis of SC gene expression. Even among SCs, however, this approach cannot resolve the differences between myelinating and non-myelinating (also known as Remak) SCs, which are expected to express different subsets of transcripts (*Gomez-Sanchez et al., 2017*; *Harty and Monk, 2017*; *Jessen et al., 2015*; *Monk et al., 2015*; *Salzer, 2015*). Studies aimed at analyzing SCs more specifically (*Clements et al., 2017*), and also the transcriptomes of other cell types in the nerve with single-cell RNA sequencing (*Carr et al., 2019*), have been largely restricted to either early embryonic development (*Buchstaller et al., 2004*; *Furlan et al., 2017*) or adult stages mainly in the context of tissue regeneration after injury (*Carr et al., 2019*; *Clements et al., 2017*).

Taken together, the rather fragmentary time points employed, together with some technical constrains, limit the universal use of the currently available data in a satisfactory manner. Thus, motivated by the demands of current research on peripheral nerve biology in health and disease, together with appreciating the highly informative nature of analogous work in the CNS (*Marques et al., 2018*; *Marques et al., 2016*; *Zhang et al., 2014*), we reasoned that a focused, easily accessible, and integrated transcriptome analysis for developing peripheral nerves and their constituent cell types would be valuable. Exemplary potential benefits include: (1) Straight-forward determination of transcript expression and changes during development and in the adult nerve, in bulk and down to the cellular level; (2) Potential use of gene expression in individual cells, or in group of cells, to expand and refine the current set of available markers that can be employed to define cells with particular structures and functions in the nerve tissue under different conditions in health and disease; (3) Identification of particularly regulated genes that may provide the basis for developing improved tools for cell-type specific labeling and genetic alteration studies; and (4) Refinement of the interpretation and integration of previous research results that were obtained with lower spatial and/or temporal cellular resolution, or with different experimental emphasis.

Thus, we performed a transcriptomic evaluation of mouse peripheral nerves using a layered approach. First, we analyzed the transcriptome of mouse peripheral nerves by bulk RNA sequencing, with a particular focus on the endoneurium compartment, covering a time-line that represents successive stages of nerve development. Second, we developed a strategy that allowed the distinction of two fractions of SCs from postnatal day 5 (P5) sciatic nerves by FACS, one enriched in myelinating SCs and the another enriched in not-myelinating SCs. Both fractions were subjected to bulk RNA sequencing. Third, we carried out single-cell RNA sequencing analysis of SCs specifically, collected by FACS at various stages of postnatal differentiation, following the Smart-Seq2 protocol. This approach enabled the recapitulation of the myelinating and non-myelinating differentiation trajectories from early development to adult stages at the molecular level. Finally, we used 10x Genomics to extend the single-cell RNA sequencing analysis to the different cell types residing in sciatic nerves, both in perinatal development and in the adult. To simplify comparative analyses, interpretations, and data mining, we merged our datasets in an openly accessible web-based resource called *Sciatic Nerve ATlas (SNAT)* at https://www.snat.ethz.ch/.

## Results and discussion

### Dynamic transcriptional changes during maturation of sciatic nerves as determined by bulk RNA sequencing

We started by resolving the transcriptomes of peripheral nerves at various time points of nerve maturation and SC differentiation using bulk RNA sequencing technology. Our analysis was largely focused on the sciatic nerve due to its wide use in studies of peripheral nerves and in SC biology in health, disease, and regeneration (*Beirowski et al., 2011*; *Fazal et al., 2017*; *Figlia et al., 2017*; *Florio et al., 2018*; *Gerber et al., 2019*; *Ghidinelli et al., 2017*; *Grove et al., 2007*; *Novak et al., 2011*; *Petersen et al., 2015*; *Quintes et al., 2016*). To account for biological diversity, we extracted nerves from different animals resulting in two independent samples derived from male mice and two independent samples from female mice per each time point. The stages analyzed included (a) embryonic day (E) 13.5, when nerves are mainly populated by SC precursors; (b) E17.5, P1 and P5,

when immature SCs gradually sort out individual large caliber axons from axonal bundles and transiently generate pro-myelinating SCs (i.e. 1:1 relationship of large caliber axons with SCs; not myelinating), followed by the onset of myelination; (c) P14 and P24, when radial sorting is completed and the key events ongoing relate to myelin growth and maturation of non-myelinating (Remak) SCs; and (d) P60, a young adult stage in which developmental myelination is complete, and non-myelinating SCs together with small caliber axons have formed mature Remak bundles (*Figure 1A*; *Feltri et al., 2016*; *Harty and Monk, 2017*; *Jessen et al., 2015*). Note that the temporal synchronization of these morphologically recognizable characteristics during peripheral nerve development is only partial, resulting in overlapping features, particularly in early postnatal development. On a relevant technical remark, we removed mechanically the epineurium and perineurium layers as much and as careful as possible on samples from E17.5 onwards to extract RNA enriched for the endoneurium compartment (*Jessen et al., 2015*; *Stierli et al., 2019*).

Standard RNA sequencing was then performed using a commercial oligo-dT-based protocol (read counts per sample are shown in *Figure 1—figure supplement 1A*). Subsequent hierarchical clustering of the samples, based on the 2000 most variable genes, revealed that the four samples per time point cluster closely together and align to recapitulate the chronological sequence of nerve differentiation (*Figure 1B*). Heat map illustrations indicated a gradual transition in the abundance of numerous mRNAs during nerve development, with high consistency between samples, also apparent in the multidimensional scaling (MDS) plot (*Figure 1—figure supplement 1B*). RPKM values for each gene are listed in *Supplementary file 1*. Given that the majority of nerve-resident cells are SCs, we decided to test the general robustness of our data sets by comparing the obtained expression levels with exemplary known expression patterns of markers that are commonly used in SC lineage analyses, in alignment to our timeline (*Figure 1C*).

First, we found as expected that the proliferation markers *Mki67* (encoding KI-67) and *Top2a* (encoding DNA Topoisomerase II Alpha) (*Carr et al., 2019*; *Xie et al., 2018*) are highly expressed from E17.5 to P5. During this time window, radial sorting of axons by SCs destined to myelinate is ongoing and SC proliferation is intense (*Jessen et al., 2015*). As maturation of the nerves proceeds, the expression of these proliferation markers decreases (*Stierli et al., 2018*). Second, we assessed the expression of markers typical of immature SCs such as *Ngfr* (encoding the Low-Affinity Neurotrophin Receptor P75NTR) (*Jessen et al., 1990*), *Ncam1* (encoding the Neural Cell Adhesion Molecule 1) and *L1cam* (encoding the L1 Neural Cell Adhesion Molecule). Peak expression was found at E17.5, consistent with the high number of immature SCs present in developing nerves at this stage (*Jessen et al., 2015*). Expression decreases thereafter, but remains substantial in agreement with the known expression of these markers also by non-myelinating SCs at later stages (*Faissner et al., 1984*; *Jessen et al., 2015*; *Martini and Schachner, 1986*; *Martini and Schachner, 1988*; *Nieke and Schachner, 1985*). Third, we detected highest expression of the pro-myelinating SC marker *Pou3f1* (encoding the Octamer-Binding Transcription Factor 6 (Oct6), also known as SCIP) (*Ghazvini et al., 2002*; *Jessen and Mirsky, 2005*; *Monuki et al., 1989*) at P1 followed by decreasing expression over time, a pattern consistent with comparative morphological analyses (*Arroyo et al., 1998*). Expression of *Cdkn1c* (encoding Cyclin-Dependent Kinase Inhibitor 1C (P57, Kip2)) (*Heinen et al., 2008*) shows a similar peak at P1, in addition to high expression at E13.5. Fourth, the levels of transcripts of the myelin protein genes *Mpz* (encoding Myelin Protein Zero), *Mbp* (encoding Myelin Basic Protein), and *Ncmap* (encoding Non-Compact Myelin-Associated Protein) (*Jessen and Mirsky, 2019*; *Martini et al., 1988*; *Patzig et al., 2011*; *Ryu et al., 2008*) increase strongly after birth, reflecting the onset of myelination.

We conclude that this dataset, obtained with bulk RNA sequencing of endoneurium-enriched nerves, provides an accurate transcriptome timeline on postnatal development of the endoneurium compartment of the mouse sciatic nerve and is suitable for appropriate data mining (see SNAT). Of technical note, it is an advantage of this approach that the SCs, as the major cell population in the nerve, together with the other parts of the endoneurium remain in their native context until immediately before RNA extraction. This feature minimizes potentially disturbing impacts of experimental procedures and optimizes that the obtained cellular RNA-seq signatures closely track genuine in vivo transcriptomes.

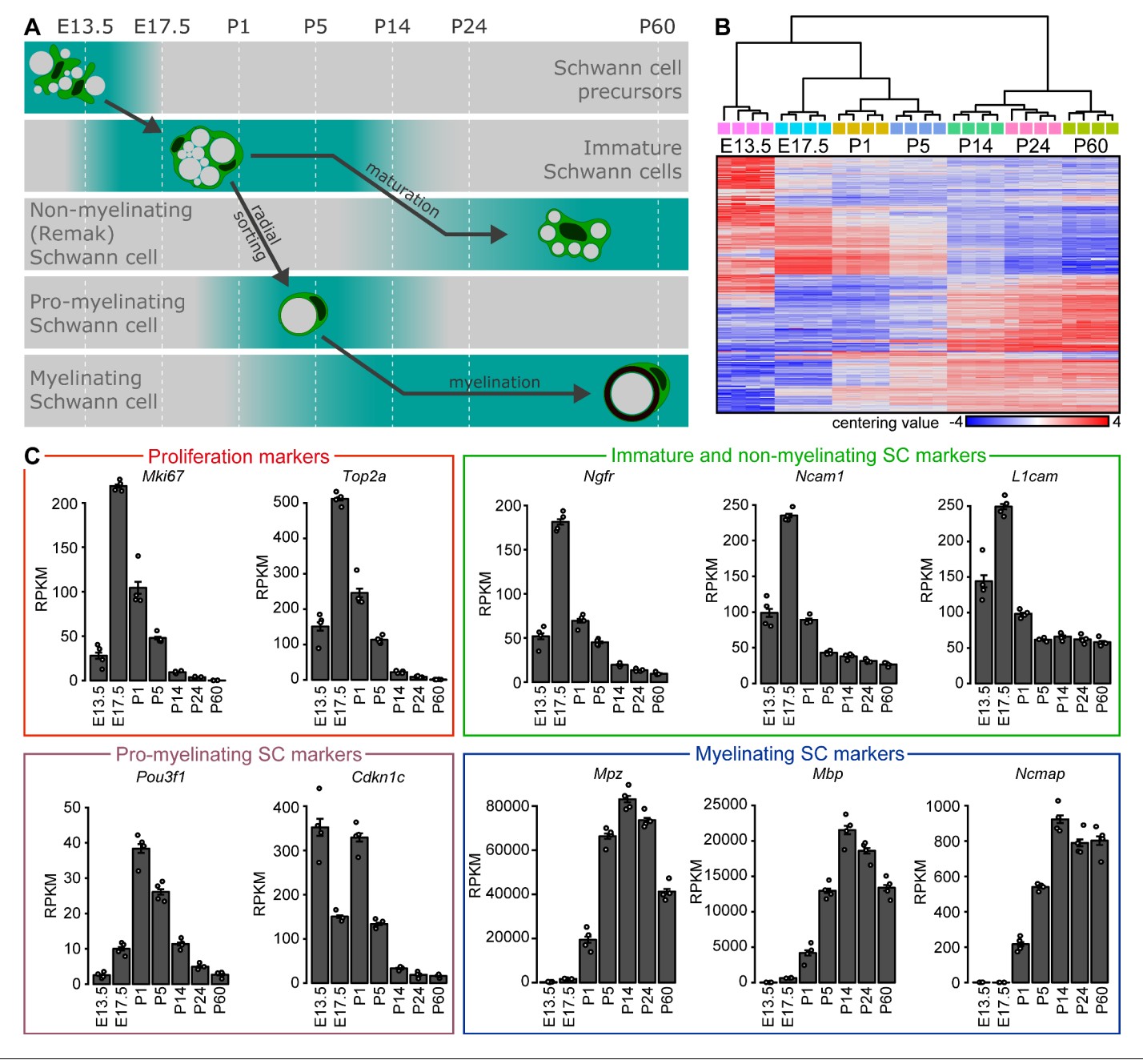

**Figure 1.** Bulk RNA profiling of peripheral nerves at various stages of embryonic and postnatal development. (**A**) Schematic representation of developmental stages used for analysis with key differentiation events of the Schwann cell (SC) lineage aligned to those stages. At embryonic day (**E**) 13.5, SC precursors can be found in peripheral nerves. At E17.5, sciatic nerves contain immature SCs which proliferate and perform radial sorting of axons. Radial sorting continues at P1 and P5. Sorted axons are engaged in a 1:1 relation by a pro-myelinating SC. These structures are frequent in sciatic nerves at P1 and P5, and to a much lesser extent at P14. Pro-myelinating SCs undergo transcriptional changes to promote the onset of myelination, followed by radial and axial growth of the myelin sheath which keeps pace with body growth at P14 and P24. Small-caliber axons not sorted for myelination remain engaged by SCs that are undergoing maturation toward a non-myelinating SC (also known as a Remak cell) at both P14 and P24. By P60, the developmental myelination program is largely completed, and sciatic nerves are widely populated by mature myelinating SCs and by mature non-myelinating (Remak) SCs. (**B, C**) Four independent samples were used for bulk RNA sequencing at every time point, two derived from male and two from female mice. Each independent sample was derived from one or two mice pooled together. At E13.5, various peripheral nerves distal to the DRG were taken from each embryo, and the nerves from two separate embryos of the same sex were pooled per sample. At E17.5, sciatic nerves were pooled from two embryos of the same sex per sample. At all postnatal time points (P1, P5, P14, P24, and P60), sciatic nerves from one mouse were used for each independent sample. (**B**) Heat map of centering values (normalized count row value of each sample subtracted by the mean

*Figure 1 continued on next page*

*Figure 1 continued*

of the row) and hierarchical cluster analysis of 2000 most variable genes during postnatal nerve development. The four independent samples sequenced at each time point cluster closer to each other than the samples of adjacent time points. Clustering of the samples reconstruct the correct chronology of nerve maturation and SC differentiation. The heat map color gradient also shows a high degree of consistency between the individual independent samples sequenced at each time point. (C) Plotting the transcript levels (RPKM) of genes related to proliferation and Schwann cell maturation throughout the timeline of analysis reveals the expected pattern of relative abundance according to the chronology of nerve maturation. The proliferation-related transcripts *Mki67* and *Top2a* are highest at E17.5 and P1. The transcripts for *Ngfr*, *Ncam1*, and *L1cam* are enriched in immature Schwann cells and peak at E17.5. These transcripts are also expressed by non-myelinating (Remak) cells, and are therefore also detected (albeit at lower levels) later in development. *Pou3f1* transcripts are enriched in pro-myelinating Schwann cells, and peak at P1 and P5. Except for E13.5, *Cdkn1c* shows a similar transcription abundance dynamic as *Pou3f1*, consistent with expression by pro-myelinating Schwann cells. Transcripts for the myelin proteins *Mpz*, *Mbp*, and *Ncmap* are accumulated from P5 onwards, around the onset of myelination, and highlight the presence of myelinating Schwann cells. See also *Figure 1—figure supplement 1*.

The online version of this article includes the following figure supplement(s) for figure 1:

**Figure supplement 1.** Library depth and MDS plot for each sample used in bulk RNA sequencing of peripheral nerves during development.

## Schwann cells isolated from sciatic nerves of P5 mice can be gated by FACS to enrich for myelinating and not-myelinating cells, allowing targeted RNA sequencing

Next, we aimed at establishing the transcriptome of developing SCs specifically, without potentially confounding contributions by other nerve-inhabiting cell types (*Carr et al., 2019*). To achieve this goal, Fluorescence-Activated Cell Sorting (FACS) was exploited to obtain purified SCs, followed by bulk RNA sequencing. In particular, we made use of transgenic mice in which SCs were labeled with the eYFP reporter protein (*Srinivas et al., 2001*) in a P0Cre-dependent manner (*Feltri et al., 1999*). For consistency with further experiments, the mice expressed also the dsRed reporter protein under the ubiquitously active chicken beta actin promoter (*Vintersten et al., 2004*). Within the nerves of these animals, SCs express both dsRed and eYFP, while other nerve-resident cells express dsRed only (*Figure 2A*). Debris and ruptured cells had low signal levels of eYFP and dsRed and were discarded during the FACS procedure (*Figure 2A*). Sciatic nerves were extracted from P5 mice and a cell suspension was generated by enzymatic digestion. Analysis by FACS revealed two semi-distinct scatter clouds of eYFP-positive SCs which can be distinguished by different degrees of granularity (SSC-A) values (*Figure 2B*). We hypothesized that these two populations may represent myelinating and not-myelinating SCs, speculating that the more granular population might relate to myelinating SCs in the nerve, and the less granular population might be enriched in SCs that were not-myelinating at the time of extraction. To evaluate this reasoning, we sorted the cells with a three-gate strategy. The first gate included all eYFP-positive SCs (marked allSC), while the second gate selected for assumed not-myelinating SCs (low SSC-A; marked nmSC), and the third gate selected for assumed myelinating SCs (high SSC-A; marked mSC) (*Figure 2A,B*). Once collected, RNA was purified from each of the three sets of cells and bulk RNA-seq analysis was performed using the Smart-Seq2 protocol. Reads per sample are shown in *Figure 2—figure supplement 1A*. The resulting RPKM values for each gene are shown in *Supplementary file 1*. To account for biological variability, we included four independent samples per gating strategy, two samples derived from female mice and two from male mice. MDS plot is shown in *Figure 2—figure supplement 1B*. Heat map and hierarchical clustering analysis revealed high consistency among the differentially regulated transcripts between samples and close clustering of individual replicates within each gating strategy (*Figure 2—figure supplement 1C*). Next, we aligned the expression levels of the exemplary genes already employed in *Figure 1*, which represent key stages of SC progression in development. The proliferation markers *Mki67* and *Top2a* are strongly elevated in the nmSC gate cells compared to the mSC gate population (*Figure 2C*), consistent with increased abundance of proliferating immature SCs in the nmSC gate population. Expression analysis of the immature SC markers *Ngfr*, *Ncam1*, and *L1cam* confirms the strong enrichment of immature SCs in the nmSC gate relative to the mSC gate (*Figure 2C*). Since the pro-myelinating SC stage represents a transient point in the SC lineage, it is difficult to predict in which gate these cells might be enriched. In this context, we found expression of *Pou3f1* to be elevated in cells of the nmSC gate, but still also to be rather substantial in cells of the mSC gate. Transcripts of *Cdkn1c* appear quite selectively enriched in cells of the nmSC gate (*Figure 2C*).

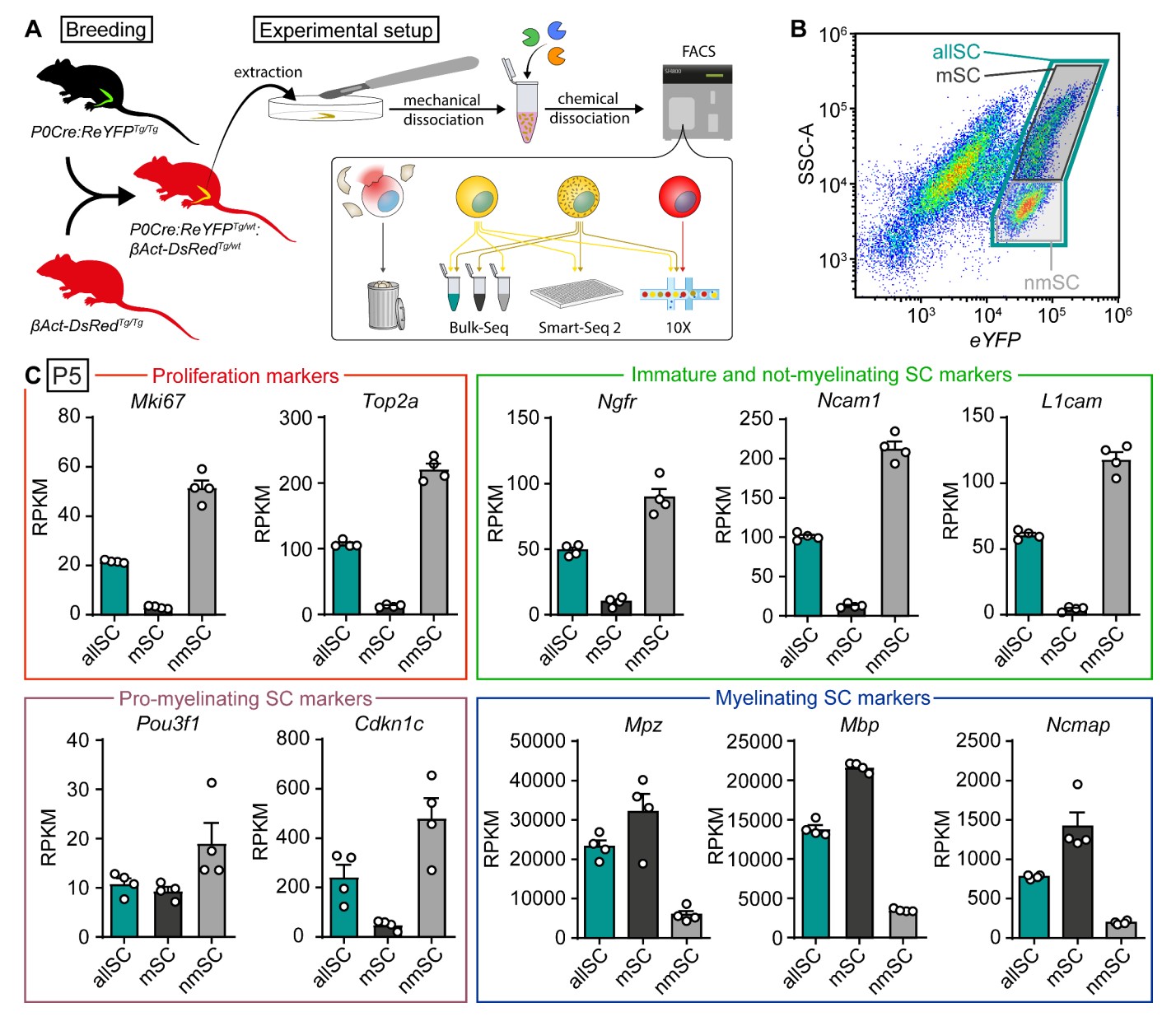

**Figure 2.** Bulk RNA profiling of sorted Schwann cells as a whole population or selectively enriched in myelinating and not-myelinating fractions. (A) Schematic representation of the general experimental setup used in the study, including the animal models expressing fluorescent reporter proteins. P0Cre:ReYFP[Tg] (schematically represented as green) labels Schwann cells (SCs) specifically with YFP, whereas the βAct-DsRed[Tg] (schematically represented as red) labels all cells with dsRed. In experimental mice, SCs in sciatic nerves express both dsRed and YFP (schematically colored in yellow), whereas other nerve cells express only dsRed (schematically colored in red). Sciatic nerves were extracted at various ages, and cell suspensions were prepared through mechanical and enzymatic dissociation. This suspension was loaded into a FACS machine, which sorted SCs based on fluorescence and granularity, depending on the experiment. Debris and ruptured cells had low levels of dsRed signal, and were therefore discarded. In the schematic, not-myelinating Schwann cells are represented in yellow, whereas myelinating Schwann cells are represented in yellow with granules. Granularity differences allowed enrichment of SC populations with lower or higher SSC-A values (see panel B). The bulk RNA sequencing shown in panels (B) and (C) used this strategy to obtain fractions enriched in myelinated (mSC) and not-myelinated (nmSC) SCs, plus a third condition of all SCs (allSC) mixed from P5 sciatic nerves. The single-cell RNA sequencing experiments shown in *Figures 3–6*, with respective supplements, carried out using Smart-Seq2 or 10x Genomics, included a mixture of myelinating and not-myelinating SCs. Non-Schwann cells in the nerve, labeled with only dsRed, were sequenced in experiments using 10x Genomics (*Figures 5* and *6*, with respective supplements). (B) Scatter plot from FACS experiments depicting YFP fluorescence levels in the x-axis, and side/back scatter (SSC-A) in the y-axis. SCs are located in the plot region related to higher YFP levels. In this region, SCs extracted from P5 nerves form two semi-distinct clouds of dots. A system of three gates were used to collect SCs. (1) a gate for SCs with higher SSC-A levels, enriched in myelinating SCs (mSC, color-coded in black frame), (2) a gate for SCs with the lower SSC-A levels, enriched in not-

*Figure 2 continued on next page*

*Figure 2 continued*

myelinating SCs (nmSC, color-coded in gray frame), and (3) a gate including both clouds (allSC, color-coded in green frame). The color-code of these gates is recapitulated in the bulk-RNA-seq schematic shown in (panel A), and in the bar graphs in (C). (C) Plotting marker transcript levels (RPKM) of genes related to proliferation and SC maturation (same markers as depicted in *Figure 1*) in SCs collected with each of the three gating strategies. In each gate, four independent replicate samples were collected, two from female mice and two from male mice. Each independent replicate consists of at least 10,000 SCs. Transcripts from the proliferation-related genes *Mki-67* and *Top2a* are strongly enriched in the nmSC gate compared to the mSC gate. Transcripts of *Ngfr*, *Ncam1*, and *L1cam*, characteristic of immature Schwann cells, are also strongly elevated within the nmSC gate relative to the mSC gate. The transcript of *Pou3f1*, expressed by pro-myelinating SCs, appears slightly enriched in the nmSC gate, even though the levels between the mSC and allSC populations are similar. Furthermore, the levels of *Cdkn1c* are also consistently higher in the nmSC compared to the mSC gate. As the pro-myelinating stage represents a transition between not-myelinating and myelinating SCs, the fine expression pattern of this gene is likely to revolve around major morphological changes in SCs, which makes it difficult to enrich for pro-myelinating SCs in this experimental setting. Transcript levels of *Mpz*, *Mbp*, and *Ncmap*, typical markers of myelinating SCs, are enriched in the mSC gate compared to the nmSC gate. As expected, the levels of most transcripts on the allSC gate were intermediate between the levels detected in the mSCs and the nmSCs gates. See also *Figure 2— figure supplement 1*.

The online version of this article includes the following figure supplement(s) for figure 2:

**Figure supplement 1.** Library depth, MDS plot, and heat map projection of bulk-sequenced Schwann cell populations at P5.

Finally, plotting mRNA levels of the myelin genes *Mpz*, *Mbp*, and *Ncmap* revealed the expected enrichment in cells of the mSC gate compared to the nmSC gate (*Figure 2C*).

Taken together, these data show that our FACS strategy allows the purification of SCs from early postnatal nerves and further enables the collection of two populations enriched in myelinating and not-myelinating cells. The possibility of such a separation and obtaining the associated RNA-seq signatures are relevant due to the major transcriptional differences between these two populations of SCs, one that has already activated the myelination program versus the other which contains SCs that are still performing radial sorting of axons (*Feltri et al., 2016*; *Jessen and Mirsky, 2005*; *Jessen and Mirsky, 2008*; *Quintes et al., 2016*). Without such a splitting, the RNA-seq data reflect an average signal between the two populations as seen in the allSC gating strategy that includes all eYFP-positive SCs (*Figure 2B,C*), yielding a convoluted picture of SC transcription dynamics around the onset of myelination. We anticipate that the described experimental strategy will be particularly valuable in the evaluation of the molecular phenotypes of mouse mutants that affect axonal radial sorting by SCs and the onset of myelination (*Figlia et al., 2017*; *Gerber et al., 2019*; *Grove and Brophy, 2014*; *Nodari et al., 2007*; *Ommer et al., 2019*; *Porrello et al., 2014*). Furthermore, our RNA-seq data of the different FACS-selected SC populations expand significantly the transcriptome resource SNAT for SC-targeted data mining.

## Single-cell RNA sequencing of Schwann cells reconstructs transcriptional fingerprints of the myelinating and non-myelinating developmental trajectories

To complement the bulk RNA sequencing data, we used single-cell RNA sequencing to determine the transcriptome of SCs across specific stages of SC differentiation, both in myelinating and non-myelinating fates, and at various time points from early development to young adults (P1, P5, P14, P60). To this end, we generated cell suspensions from mouse sciatic nerves and used the 'allSC' FACS gate strategy depicted in *Figure 2B*. Single eYFP-positive SCs were sorted into 384-well plates, followed by single-cell RNA sequencing with the Smart-Seq2 protocol (*Figure 2A*). Biological and technical variability was accounted for by preparing two plates per age. Each plate contained sorted cells derived from different animals. The single-cell RNA sequencing data were projected as tSNE plots. Analysis of color-coded individual cells per plate of origin (designated plate 1 and plate 2, per time point) indicated homogeneous distribution of cells derived from both plates at P1, P5, and P14 (*Figure 3—figure supplement 1B*). At P60, a shift in cell distributions according to the origin of the plate was apparent (*Figure 3—figure supplement 1C*). Thus, we applied batch correction to this dataset to minimize the bias (*Figure 3—figure supplement 1B*, *Figure 3G*). Further analysis revealed three cell clusters at P1, and four clusters at P5, P14, and P60, which are individually color-coded on tSNE projections per time point in (*Figure 3A,C,E,G*). We detected a proliferating SC cluster (marked prol. SC) at P1, P5 and P14, containing cells that express *Mki67* and *Top2a* (*Figure 3B, D,F*). Some SCs expressing these markers were also found in the non-myelinating (Remak) cell

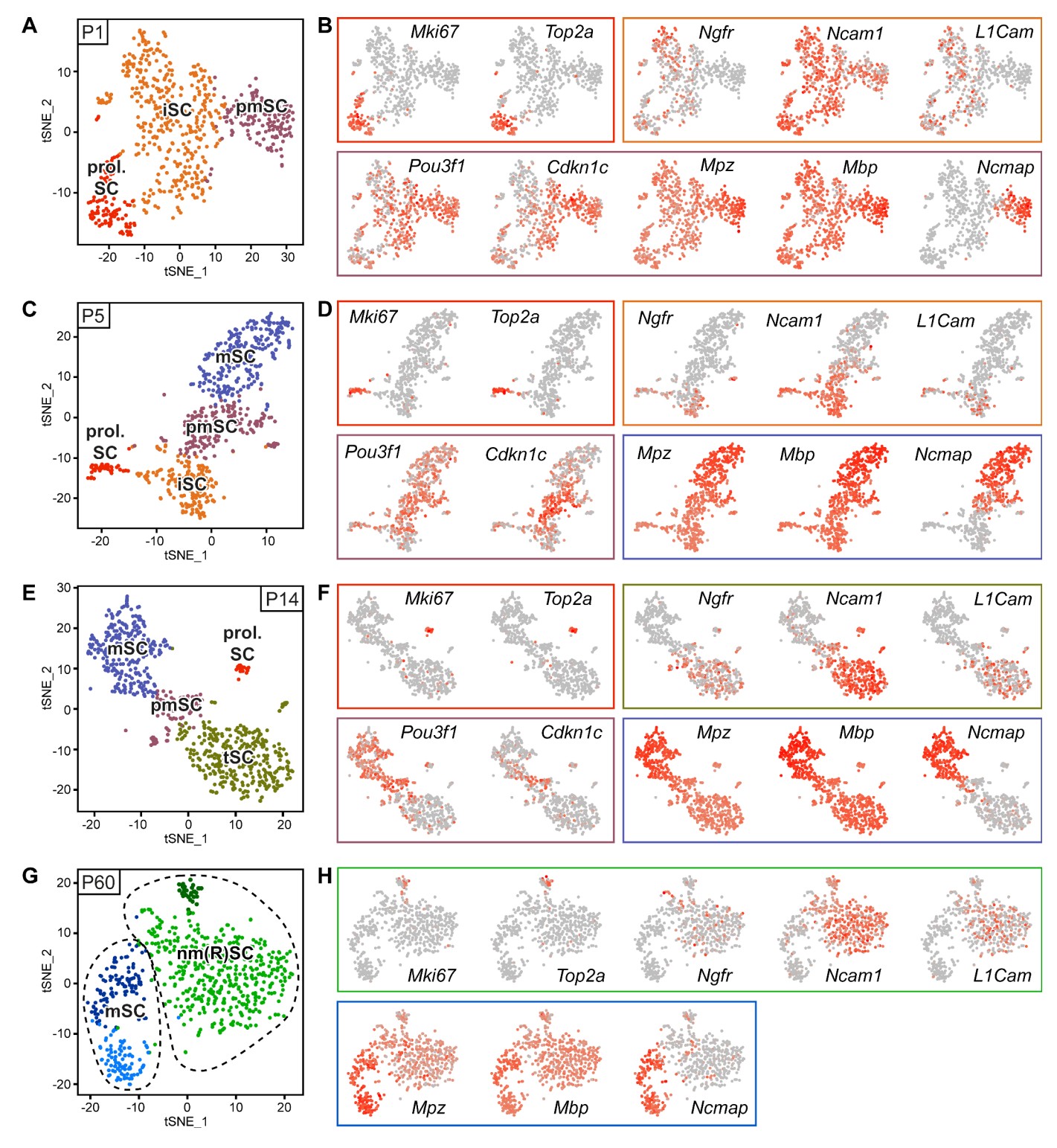

**Figure 3.** RNA single-cell profiling of Schwann cells handled with Smart-Seq2 protocol over various stages of postnatal development. (**A–H**) Cells extracted from sciatic nerves were sorted based on expression of YFP (selection for Schwann cells (SCs)). Single YFP-positive SCs were individually sorted into single wells of a 384-well plate, containing lysis reagents for Smart-Seq2 library preparation (see schematic in *Figure 2A*). Four postnatal time points were selected, P1, P5, P14, and P60, covering various stages of SC differentiation. Two 384-well plates, each plate containing cells derived from different animals, were sorted and subjected to sequencing at each time point. Cells from each plate derived from a cell-suspension extracted from male and female mice in a 1:1 ratio. One male and one female mouse were pooled per plate at P1, P5, and P14, while two male and two female

*Figure 3 continued on next page*

*Figure 3 continued*

mice were pooled per plate at P60. (**A**) tSNE plot of SCs extracted from sciatic nerves of P1 mice. Data analysis indicates three distinct clusters, one relating to proliferating Schwann cells (prol. SC), another to immature Schwann cells (iSC), and a third cluster to pro-myelinating Schwann cells (pmSC). (**B, D, F, H**) tSNE plots with color overlays indicating transcript levels per cell. A gradient of red tones indicates relative abundance of each transcript per cell, in which dark red indicates higher levels, and gray indicates not detected. The color of the surrounding frame relates to the color-code of each cluster. Markers expressions described are enriched in (although not exclusive to) the cluster indicated. (**B**) The proliferation transcripts *Mki67* and *Top2a* label the prol. SC cluster. *Ngfr*, *Ncam1*, and *L1cam* are enriched in the iSC clusters. *Pou3f1* and *Cdkn1c* are slightly enriched in the pmSC cluster. The myelin protein transcripts *Mpz*, *Mbp*, and *Ncmap* are particularly enriched in the pmSC cluster. (**C**) tSNE plot of SCs extracted from P5 mice. Data analysis indicates four distinct clusters, one relating to prol. SC, another to iSC, a third cluster to pmSC, and a fourth cluster relates to myelinating SCs (mSC). (**D**) *Mki67* and *Top2a* localize to the prol. SC cluster; *Ngfr*, *Ncam1* and *L1cam* are enriched in the iSC cluster; and *Pou3f1* and *Cdkn1c* are enriched in the pmSC cluster. The myelin protein transcripts *Mpz*, *Mbp*, and *Ncmap* are accumulated in the mSC cluster. (**E**) tSNE plot of SCs extracted from P14 mice. Data analysis indicates four distinct clusters, one relating to prol. SC, another to non-myelinating cells undergoing maturation and marked here as transition SCs (tSC), a third cluster to pmSC, and a fourth cluster depicts mSC. (**F**) As expected, *Mki67* and *Top2a* localize to the prol. SC cluster; *Ngfr*, *Ncam1*, and *L1cam* are enriched in the tSC cluster; and *Pou3f1* and *Cdkn1c* are enriched in the pmSC cluster. The myelin protein transcripts *Mpz* and *Mbp* and *Ncmap* are enriched in the mSC cluster. (**G**) Batch-corrected tSNE plot of SCs extracted from P60 mice. Data analysis indicates four distinct clusters, with two clusters relating to mature non-myelinating (Remak) SCs (nm(R)SC), and two clusters related to myelinating SCs. (**H**) *Mki67* and *Top2a* are proliferation-related genes labeling some cells in the nm(R)SC clusters; *Ngfr*, *Ncam1* and *L1cam* mainly localize to the nm(R)SC clusters. The myelin protein transcripts *Mpz*, *Mbp*, and Ncmap are enriched in the mSC clusters. Dashed lines indicate a merge of clusters produced in the default output of the analysis (*Figure 3—figure supplement 2*). See also *Figure 3—figure supplement 1*.

The online version of this article includes the following figure supplement(s) for figure 3:

**Figure supplement 1.** tSNE plots depicting the source plate of each cell, and gene detection depth per cell in Schwann cells handled with Smart-Seq2 through different stages of postnatal development.

**Figure supplement 2.** Original clusters overlaid with tSNE plots related to single-cell sequencing analysis of sciatic nerve cells handled with Smart-Seq2 at individual time points.

clusters (marked nm(R)SC) at P60, consistent with previous findings (*Stierli et al., 2018*; *Figure 3G, H*). Immature SCs (marked iSC) also formed a cluster at P1 and P5, characterized by expression of *Ngfr*, Ncam1, *and L1cam* (*Figure 3B,D*). The same markers are also enriched in Remak SCs undergoing maturation at P14 (*Figure 3F*), designated as transition SCs (marked tSC), and as mature Remak SCs (marked nm(R)SC) at P60 (*Figure 3H*) (Note that the designation 'tSC' is not intended to present SCs at P14 as a discrete cell type. In our interpretation, tSC refers rather to Remak SCs which are likely undergoing maturation). A cluster of pro-myelinating SCs (marked pmSCs) could be detected at P1, P5, and P14, with enriched expression of *Pou3f1* and *Cdkn1c* (*Figure 3B,D,F*). At P1, this cluster also contains SCs expressing *Mpz*, *Mbp*, and *Ncmap* (*Figure 3B*), while cells expressing these transcripts at later time points (P5, P14, and P60) are strongly enriched in myelinating SC clusters (marked mSC) (*Figure 3D,F,H*). At P60, data processing suggested two clusters labeled by myelinating SC markers and two clusters labeled by non-myelinating (Remak) SC markers (*Figure 3G,H*). We grouped each of these cluster pairs together as myelinating SCs (marked mSC) and non-myelinating (Remak) SCs (marked nm(R)SC), respectively. Gene expression detection depth per cell is given in *Figure 3—figure supplement 1A*. The default output of cluster formation in the datasets is provided in *Figure 3—figure supplement 2*.

Individual evaluations of the datasets per time point depict the clusters for the main expected differentiation stages of SCs, but in a fragmentary manner. To obtain a more integrated perspective of transcriptomic dynamics over postnatal SC differentiation, we combined all data obtained by single-cell SC sequencing in an integrated analysis. This strategy resulted in one tSNE plot of cells from all time points (*Figure 4A*). Mapping the cellular transcript expression of selected differentiation marker genes within the combined-data plot revealed a close association of clusters with known key stages of SC differentiation (*Figure 4B*). Proliferating SCs (marked prol. SC) form a cluster marked by expression of *Mki67* and *Top2a*. The iSC cluster is marked by expression of *Ngfr*, *Ncam1*, and *L1cam*. The same markers also label tSC and nm(R)SC. As a group, these clusters and markers describe the known differentiation and maturation path of non-myelinating (Remak) SCs (*Figure 4B*). A second group involves also proliferating SCs and the iSC cluster (which are common to both groups), the pmSC cluster enriched in *Pou3f1* and *Cdkn1c* expression, and the SC clusters with strong enrichments in transcripts of *Mpz*, *Mbp*, and *Ncmap* (marked mSC) (*Figure 4B*, *Figure 4—figure supplement 1*). Together, these clusters and markers describe the known differentiation and maturation path of myelinating SCs. Color-coding individual cells according to the age of origin

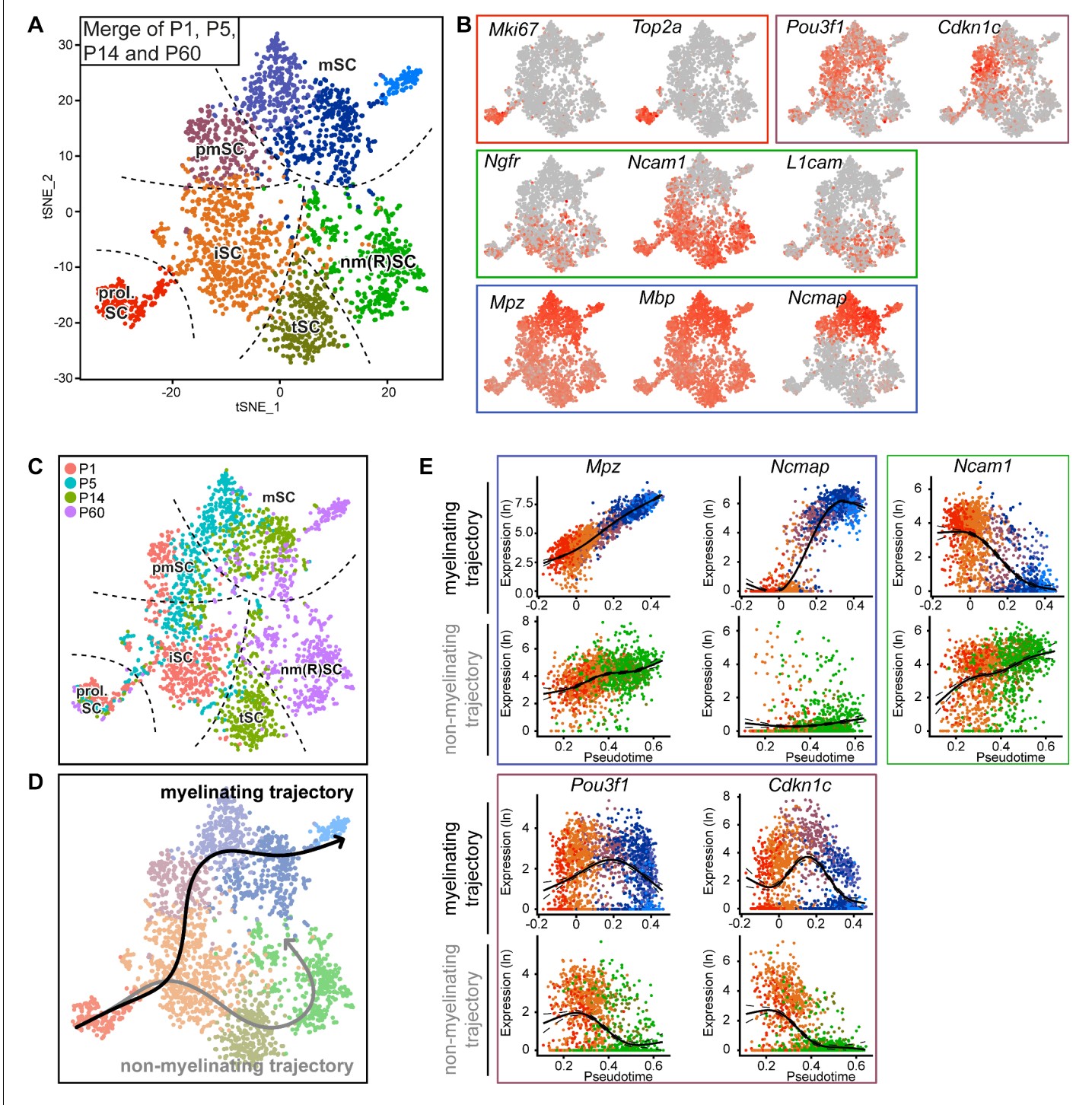

**Figure 4.** Integrated RNA profiling with single-cell resolution covering various stages of Schwann cell development. (**A**) tSNE plot derived from single-cell RNA sequencing of Schwann cells (SCs) handled with the Smart-Seq2 protocol. All data presented in *Figure 3* were condensed into a single plot. Data analysis resulted in eight clusters, marked with the same nomenclature as in *Figure 3*. These clusters include proliferating SCs (prol. SC), immature SCs (iSC), transition SCs (tSC), mature non-myelinating (Remak) SCs (nm(R)SC), pro-myelinating SCs (pmSC), and myelinating SCs (mSC). (**B**) tSNE plot with color overlay indicating detected transcript levels per cell. A gradient of red tones indicates relative abundance of each transcript per cell, in which dark red indicates higher levels, and gray indicates not detected. The color of the surrounding frame relates to the color code of each cluster. The markers expressions described here are enriched in (although not exclusive to) the cluster indicated. Expression of *Mki67* and *Top2a* was enriched in the prol. SC cluster. The levels of *Ngfr*, *Ncam1*, and *L1cam* were enriched in the iSC, tSC and nm(R)SC clusters. *Pou3f1* and *Cdkn1c* were enriched in the pmSC cluster. Transcripts for the myelin proteins *Mpz*, *Mbp*, and *Ncmap* are high in the myelinating SC clusters (mSC). (**C**) tSNE plot including all

Figure 4 continued

Schwann cells (SCs) from all ages handled with the Smart-Seq2 protocol, with cells color-coded to their age of analysis (P1, P5, P14, and P60). The distribution of the cells with increasing age relates to the differentiation trajectories predicted in (D) and is congruent with known morphological features observed in the nerve at the different ages. Dashed lines outline the approximate region of each cluster. (D) Smooth curves reconstructed by supervised application of the Slingshot package depict two branching trajectories, with the prol. SC set as starting cluster. One of the branches depicts the myelinating trajectory (black arrow, top), and the other branch depicts the non-myelinating trajectory of SC maturation (gray arrow, bottom). (E) Temporally changing gene expression based on the two trajectories produced by Slingshot shown in (D). In each frame, the top row depicts gene expression change along the myelinating trajectory, whereas the bottom row shows gene expression change along the non-myelinating trajectory. Within the blue frame, transcripts of *Mpz* and *Ncmap* strongly accumulate over pseudotime in the myelinating trajectory but not in the non-myelinating trajectory. Within the green frame, transcripts of *Ncam1* are reduced over pseudotime along the myelinating trajectory, and increased over pseudotime along the non-myelinating trajectory. Within the purple frame, transcript levels of *Pou3f1* and *Cdkn1c* peak at the pro-myelinating stage and are lower before and after in pseudotime along the myelinating trajectory, while both are decreasing along pseudotime in the non-myelinating trajectory. See also *Figure 4—figure supplement 1* and *Figure 4—figure supplement 2*.

The online version of this article includes the following figure supplement(s) for figure 4:

**Figure supplement 1.** Original clusters overlaid with the tSNE plot related to single-cell sequencing analysis of sciatic nerve cells handled with Smart-Seq2 covering the various integrated time points.

**Figure supplement 2.** Integrated tSNE plot of Schwann cells handled with Smart-Seq2 depicting the gene detection depth per cell.

generated a chronological gradient over the tSNE plot that agrees with the pattern of overlapped marker transcripts and the two differentiation paths (*Figure 4C*) (compare also to the description of individual time points in *Figure 3*). SCs derived from nerves of younger animals correspond to earlier stages of cell differentiation than those obtained from older mice (*Figure 4C*). On a discussion note, the putative non-myelinating SC maturation path includes two informatic clusters, one enriched in P14 SCs (marked tSC) and the other in P60 SCs (marked nm(R)SC). These findings may reflect temporal snapshots of the maturation fate of non-myelinating (Remak) SCs. Likewise, the informatically predicted mSC clusters are differentially enriched in mSCs of progressing mouse ages, potentially reflecting the particular requirements of early and young adult phases of the myelination program. To aid visualizing the development of transcript levels along the two SC differentiation and maturation trajectories as suggested by the combined-data tSNE plot, we applied a supervised Slingshot trajectory and pseudotime inference approach (*Street et al., 2018*). Proliferating SCs were defined as the starting cluster, and trajectories crossed the center of each cluster. As expected, the results of this approach indicated a bifurcated projection that splits in two major branches after the iSC cluster, one leading toward the myelinating fate and the other toward non-myelinating (Remak) SC specification (*Figure 4D*) (Note that the analysis was not aimed at identifying novel cell differentiation states nor trajectories, but is limited to the visualization of results obtained by a supervised bioinformatic approach in conjunction with current knowledge in the research field). The pseudotime analysis allowed the (re-)creation of a virtual timeline by associating each cell with a temporal variable along each path of the transcriptional changes (*Figure 4E*). For example as expected, transcripts of *Ncam1* get strongly reduced with differentiation on the myelinating trajectory, but remain high along the non-myelinating trajectory. The transcripts *Pou3f1* and *Cdkn1c* are prominently enriched along the myelinating trajectory in a transient manner (*Figure 4E*). Furthermore, transcripts for *Mpz* are accumulated along the myelinating trajectory, albeit also detected at lower levels in all other SC clusters. In comparison, transcripts for *Ncmap* are more restricted to late stages of the myelinating trajectory (*Figure 4E*). In addition to the already presented selected markers related to specific stages of SC differentiation, further information is provided by the enriched transcripts in each cluster of the tSNE plot in *Supplementary file 3*. Additional technical information regarding the sequencing data for individual cells of the tSNE projection, including the total number of detected genes, is provided in *Figure 4—figure supplement 2A*, together with the number of reads mapped to the transcriptome per cell (nCount) in *Figure 4—figure supplement 2B*, and the fraction of mitochondrial content per cell in *Figure 4—figure supplement 2C*.

Taken together, our Smart-Seq2 single-cell RNA sequencing dataset provides a connected collection of clusters along key stages of SC postnatal differentiation. Furthermore, the dataset enables a direct comparison, with single-cell resolution, of the differential transcriptome changes throughout SC maturation during postnatal development. The dataset as such, together with the application of the analytical tools presented, provides the fundamental groundwork to allow detailed comparative

mapping of the expression of genes by SCs within the myelinating and non-myelinating trajectory throughout development, a prerequisite for the functional understanding of a given gene in SC differentiation and maturation. We anticipate that such applications will be fostered and facilitated particularly by the use of the accompanying database SNAT.

## Single-cell RNA sequencing analysis reveals molecular hallmarks of the major cell populations that contribute to sciatic nerve function

Various other cell types besides SCs and neurons/axons contribute together to the optimal function of peripheral nerves. However, our understanding of the individual roles of these different cell types in nerve development and homeostasis are limited. Comparatively more is known about the essential roles of some of those non-neural cells in nerve regeneration. This knowledge includes a crucial interplay between macrophages, endoneurial fibroblast-like cells, and SCs in fostering axonal regeneration between proximal and distal nerve stumps after nerve transection (*Cattin et al., 2015*; *Cattin and Lloyd, 2016*). Furthermore, nerve-resident *Pdgfra*-expressing cells (which include cells of the endoneurium, perineurium, and epineurium) have been shown to serve as a cellular reservoir during regeneration after toe tip amputation of adult mice (*Carr et al., 2019*). The same study reported also on data obtained by single-cell RNA sequencing of nerve cells from adult injured tissue and included the analysis of *Pdgfra*-expressing cells of adult uninjured nerves. We reasoned that extended and easy searchable datasets describing transcriptomic dynamics in the various resident cell types of the developing and adult sciatic nerve would provide a valuable resource, in particular if put in a consistent context with our other described sets of RNA-seq data in an integrated database. Thus, we decided to expand our single-cell RNA sequencing experiments to other sciatic nerve-inhabitant cells. To this end, we generated cell suspensions from all-inclusive sciatic nerves with the same protocol that we had used to extract SCs in the single-cell sequencing approach before. The presence of dsRed allowed the use of FACS to separate viable nerve cells, which were strongly positive for this cytoplasmic reporter, from debris and ruptured cells (*Figure 2A*). Our study includes the analysis of cells at early postnatal development, extracted from mice at P1, and cells at a young adult stage, extracted from mice at P60. The additional presence of the eYFP reporter, expressed in a P0Cre-dependent manner, allowed convenient monitoring of the SC fraction during FACS sorting. Thus, we could select negatively for SCs at P1 (reducing the SC number down to ~30% of the total cells analyzed at this age) to restrict the quantitative dominance of SCs that would have potentially hindered the analysis of other cell types in sufficient numbers for informatic cluster formation. We chose to use the 10x Genomics technology rather than Smart-seq2 for this part of our study, since higher cell throughput is required to characterize adequately the heterogeneous cell types expected in the sciatic nerve. Approximately 4000 sorted nerve cells were loaded into the Chromium controller for separation of single cells into droplets during each individual run, with cDNA and libraries prepared according to 10x Genomics recommendations (*Figure 2A*). To account for biological variability and to control for technical bias, we included three independent samples for each time point, processed as three independent 10x Genomics runs (designated as Run 1, 2, and 3). Each sample consisted of a nerve cell suspension extracted from male and female mice in a 1:1 ratio. Color-coding of the cells in the tSNE plot (*Supplementary file 2*) according to different runs revealed no major bias, as cells from all three runs are contributing to the formation of each cluster (*Figure 5—figure supplement 1A*, *Figure 6—figure supplement 1A*). To identify the clusters represented in the tSNE plots, we pinpointed cells expressing transcripts from genes that specifically label the various cell types expected in the nerve. Since the available literature validating these markers is mainly derived from adult animals, we prioritized the identifications of each cluster in P60 samples. Analysis of the P60 single-cell RNA-seq data revealed nine clusters (*Figure 5A*).

SCs accounted for 414 out of 12428 cells (3.3%; note that this number is affected by the experimental extraction efficiency and does not reflect the proportion of SCs present in adult sciatic nerves [*Stierli et al., 2018*]), formed a single cluster (marked SC) in the tSNE, and were characterized by *Sox10* (encoding the transcription factor SOX10) expression (*Finzsch et al., 2010*; *Jessen et al., 2015*; *Figure 5A,B*). However, closer inspection revealed a remarkable regional distribution of myelinating SCs (marked mSC) on one side of the cluster and non-myelinating (Remak) SCs on the opposite side (marked nm(R)SC) (*Figure 5C,D*). Expression of the myelin genes *Mpz*, *Mbp*, and *Ncmap* is enriched in the mSC region of the cluster, whereas *Ngfr* and *Ncam1* expression are enriched in the nm(R)SC area of the cluster (*Figure 5C,D*).

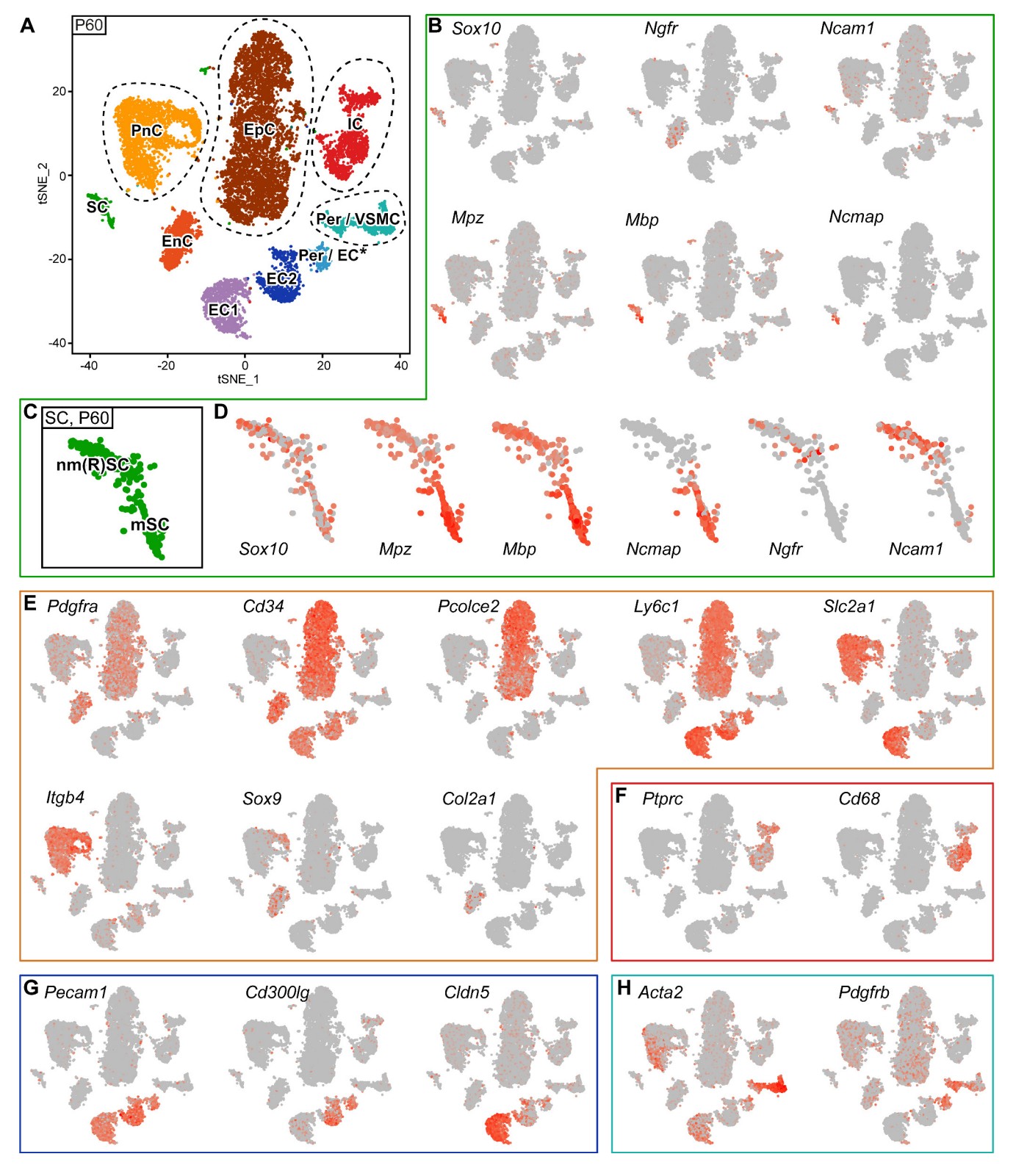

**Figure 5.** RNA single-cell profiling of sciatic nerve cells handled with 10x Genomics at P60. (**A**) tSNE plot depicting the major clusters of nerve-resident cells based on RNA sequencing of single cells extracted from sciatic nerves at P60 and handled with 10x Genomics. These plots include single clusters of Schwann cells (SC), perineurial cells (PnC), endoneurial cells (EnC), epineurial cells (EpC), and immune cells (IC). Pericytes and vascular smooth muscle cells (Per/VSMC) are represented as one cluster. A second cluster marked as pericytes/endothelial cells (Per/EC*) includes defining markers of both

*Figure 5 continued on next page*

Figure 5 continued

pericytes and endothelial cells within the same cell. The tentative nature of this cluster is indicated by an asterisk (see text). Markers typical of endothelial cells form two separate clusters, marked as endothelial cells 1 and 2 (EC1, EC2). Dashed lines indicate a merge of clusters produced from the default output of the analysis (see *Figure 5—figure supplement 3*). (B, D–H) tSNE plots overlaid with the detected relative transcript levels. A gradient of red tones indicates relative abundance of each transcript per cell, in which dark red indicates higher levels, and gray indicates not detected. The markers described label the cluster indicated, although not always exclusively. (B–D) tSNE overview in (B), and focused on Schwann cell cluster (SC) in (D), overlaid with key markers for all SCs (*Sox10*), myelinating SCs (*Mpz*, *Mbp*, and *Ncmap*), and non-myelinating SCs (*Ngfr* and *Ncam1*). The distribution pattern of these transcripts across the cluster indicates regions enriched in nm(R)SC and in mSC (C). (E) tSNE plots overlaid with the detected relative transcript levels for EnC, PnC, and EpC clusters. All three clusters are labeled by *Pdgfra*, whereas *Cd34* labels PnC at lower levels than EnC and EpC. The EpC cluster is labeled by *Pcolce2* and *Ly6c1*, the PnC cluster is labeled by *Slc2a1* and *Itgb4*, and the EnC cluster is labeled by *Sox9* and *Col2a1*. (F) The IC cluster is labeled by *Ptprc* and *Cd68*. (G) The two clusters representing endothelial cells (EC1 and EC2) are both labeled by Pecam1, as is the Per/EC cluster. The EC2 cluster is predominantly labeled by expression of the marker *Cd300lg*, whereas the EC1 cluster is labeled by the tight junction marker *Cldn5*. (H) The Per/VSMC cluster is labeled by *Acta2* and *Pdgfrb*. See also *Figure 5—figure supplement 1*, *Figure 5—figure supplement 2* and *Figure 5—figure supplement 3*.

The online version of this article includes the following figure supplement(s) for figure 5:

**Figure supplement 1.** Integrated tSNE plot of nerve cells handled with 10x Genomics at P60, encoding the chromium run that each cell derives from, the most enriched transcripts per cluster, and the gene detection depth per cell.

**Figure supplement 2.** Schematic representation of a sciatic nerve cross-section with the major cell types distributed in the different nerve layers.

**Figure supplement 3.** Original clusters overlaid with tSNE plots related to single-cell sequencing analysis of sciatic nerve cells handled with 10x Genomics at P60.

As previously shown (*Carr et al., 2019*), enriched *Pdgfra* expression labeled clusters of epineurial (marked EpC), perineurial (marked PnC), and endoneurial fibroblast-like (marked EnC) nerve cells, whereas expression of *Cd34* (encoding Hematopoietic Progenitor Cell Antigen CD34) labels all of the above (albeit the perineurial cell cluster only weakly), and additionally also endothelial cell clusters (*Carr et al., 2019*; *Richard et al., 2012*; *Richard et al., 2014*; *Figure 5A,E*). To identify each cluster more specifically, we overlaid additional marker genes on the tSNE plots. Enriched expression of *Pcolce2* (encoding Procollagen C-Endopeptidase Enhancer 2) and *Ly6c1* (encoding lymphocyte antigen 6 complex, locus C1) label the epineurial cell cluster (*Figure 5E*; *Carr et al., 2019*). The perineurial cell cluster is labeled by expression of *Itgb4* (encoding Integrin Subunit Beta 4) and *Slc2a1* (encoding Solute Carrier Family 2 Member 1, also commonly known as Glucose Transporter Type 1(Glut1)) (*Figure 5E*; *Carr et al., 2019*; *Niessen et al., 1994*; *Takebe et al., 2008*; *Tserentsoodol et al., 1999*). Finally, the endoneurial fibroblast-like cell cluster is characterized by enriched expression of *Col2a1* (encoding Collagen Type II Alpha 1 Chain) and *Sox9* (encoding the transcription factor SOX9) (*Figure 5E*; *Carr et al., 2019*).

The immune cell cluster (marked IC) can be recognized by *Ptprc* (encoding Protein Tyrosine Phosphatase Receptor Type C, also commonly known as CD45) expression and particularly high abundance of the macrophage marker *Cd68* (encoding Scavenger Receptor Class D, Member 1) (*Figure 5F*; *Carr et al., 2019*; *Cattin et al., 2015*; *Hidmark et al., 2017*).

Cells associated with the remaining clusters are related to blood vessels. Two clusters are characterized by ample expression of *Pecam1* (Platelet And Endothelial Cell Adhesion Molecule 1, also commonly known as CD31) (*Figure 5G*; *Carr et al., 2019*; *Cattin et al., 2015*; *Stierli et al., 2018*), but not of *Pdgfrb* (encoding Platelet Derived Growth Factor Receptor Beta) (*Figure 5H*). We classified these groups as bona fide endothelial cell clusters. From these two clusters, one was more enriched in *Cldn5* (encoding the tight junction component Claudin 5) (*Pummi et al., 2004*; *Figure 5G*) and *Slc2a1* (*Takebe et al., 2008*; *Tserentsoodol et al., 1999*; *Figure 5E*) expression (marked EC1). The other endothelial cluster had lower expression of *Cldn5* and *Slc2a1* (*Figure 5E,G*), together with higher abundance of the endothelial marker *Cd300lg* (encoding CD300 Molecule Like Family Member G) (*Figure 5G*; *Zhao et al., 2018*) (marked EC2). Peripheral nerves contain a complex set of blood vessels, including larger vessels located mainly in the epineurium, and microvessels within the endoneurium (*Reinhold and Rittner, 2020*; *Ubogu, 2020*). One might speculate that the two observed endothelial clusters may relate to different sets of endothelial cells, with different functional properties associated with vessels with different physical and physiological features. Next, a cluster labeled by expression of *Acta2* (encoding Actin Alpha 2, Smooth Muscle) and *Pdgfrb* was identified as pericytes and vascular smooth muscle cells (marked Per/VSMC) (*Figure 5H*; *Carr et al., 2019*; *Stierli et al., 2018*). Finally, the remaining discernible cluster of this

blood vessel-associated group contains cells that are simultaneously positive for *Pecam1* and *Pdgfrb* (*Figure 5G,H*), canonical markers for endothelial cells and pericytes, respectively (marked Per/EC*). One interpretation of this finding is that sciatic nerves may contain pericytes that also express *Pecam1*, or endothelial cells that also express *Pdgfrb*. Since we cannot exclude this possibility at this time, we did not remove the cluster from the dataset. However, we consider it more likely that this cluster derives from 'cell doublets' between endothelial cells and pericytes, and thus is a technical artefact potentially fostered by the close physical interactions between pericytes and endothelial cells on vessels, a connection that may not have been broken sufficiently by our nerve dissociation procedure.

The markers discussed above and plotted in *Figure 5* are a targeted selection based on previous knowledge. Exemplary citations associated with these markers, together with an illustrating diagram of the distribution of the major cell types within the nerve compartments in relation to the RNA sequencing, are shown in *Figure 5—figure supplement 2*. Row z-scores of the ten genes with the most enriched transcripts per cluster within the dataset are included as a heat map projection (*Figure 5—figure supplement 1B*), and a more comprehensive list is also included (*Supplementary file 3*). Additional technical information regarding the sequencing data is shown on individual cells of the tSNE projection and includes the number of detected genes (*Figure 5—figure supplement 1C*), the fraction of mitochondrial content detected (*Figure 5—figure supplement 1D*), and the number of unique molecular identifiers (*Figure 5—figure supplement 1E*).

Our complementary analysis of developing nerves at P1 resulted in a tSNE projection containing 11 clusters (*Figure 6A*). Notably, even after the previously described negative selection by FACS, the P1 SC cluster still included a larger number and fraction of SCs than obtained at P60 (3175 or 28% (P1), vs. 414 or 3.3% (P60)), likely reflecting a more efficient extraction of intact SCs from P1 nerves. Three SC clusters were identified by *Sox10* expression (*Figure 6B*). As a group, they are also highly enriched in *Mpz*, *Mbp*, and *Ncmap* transcripts. *Ncmap* expression is most prominent in a SC cluster that reflects most likely pro-myelinating SCs (marked pmSC) at this age (*Figure 6B*), although a few SCs might also be included here that have already started to produce myelin. An overlapping neighboring group, labeled by *Ncam1* and *Ngfr* expression, includes immature SCs (marked by iSC) (*Figure 6B*), some of which also expressing the proliferation markers *Mki67* and *Top2a* (marked prol. SC) (*Figure 6C*). The expression of these proliferation markers labels also other proliferating cells in P1 nerves, which we tentatively designated as a cluster of proliferating fibroblast-like cells (marked prol. Fb) due to the expression of *Pdgfra* (*Figure 6C,D*). To relate other clusters to the identity of the major cell types expected in the nerve, we employed the markers used at P60 before and plotted them in the tSNE plot of P1 nerves, with the limitation of assuming that the same markers are useful identifiers of the same cell type clusters analogously to P60 also at P1. As mentioned before, a cluster of proliferating fibroblasts was labeled by *Pdgfra* expression, as were four additional clusters in the tSNE plot also labeled by this marker. We reasoned that these clusters are likely to relate to endoneurial fibroblast-like cells (marked EnC), perineurial cells (marked PnC), epineurial cells (marked EpC), leaving one other cluster which we called tentatively fibroblast-related cells (marked FbRel*). Assignments of these clusters were done as follows: Pronounced expression of *Sox9* and *Col2a1* identified the putative EnC cluster (*Figure 6D*). Next, expression of *Ly6c1* and *Pcolce2* allowed the localization of the putative EpC cluster (*Figure 6D*). Afterwards, we hypothesized that one of the remaining *Pdgfra*-expressing clusters should comprise perineurial cells. Thus, we assigned a putative PnC cluster based on a modest enrichment of *Slc2a1* expression (*Figure 6D*). In this context, we note again that these detailed assignments have to be considered tentative at this time, since the marker genes used for the assignments at P1 are likely expressed at different levels than at P60 due to the different cell differentiation states. The fibroblast-related cluster (FbRel*) includes cells in which, besides *Pdgfra*, a peculiar set of transcripts are quite prominent, including some cells expressing markers more enriched in other clusters, and even some cells expressing genes typical of the SC lineage, including *Mbp*, *Mpz*, and *Sox10* (*Figure 6B,D*). Determination of the origins and validation of this cluster requires further investigations.

Clusters of immune cells (marked IC) were assigned by *Ptprc* expression, with a preponderance of *Cd68*-expressing macrophages similar to the findings at P60 (*Figures 6E* and *5F*). Putative endothelial cells formed a single cluster labeled by *Pecam1* and not *Pdgfrb* expression (marked EC; *Figure 6F and G*). This cluster contains some cells expressing *Cldn5* and some with *Cd300lg* expression. However, there is no evident fractionation of the cluster apparent at this early developmental

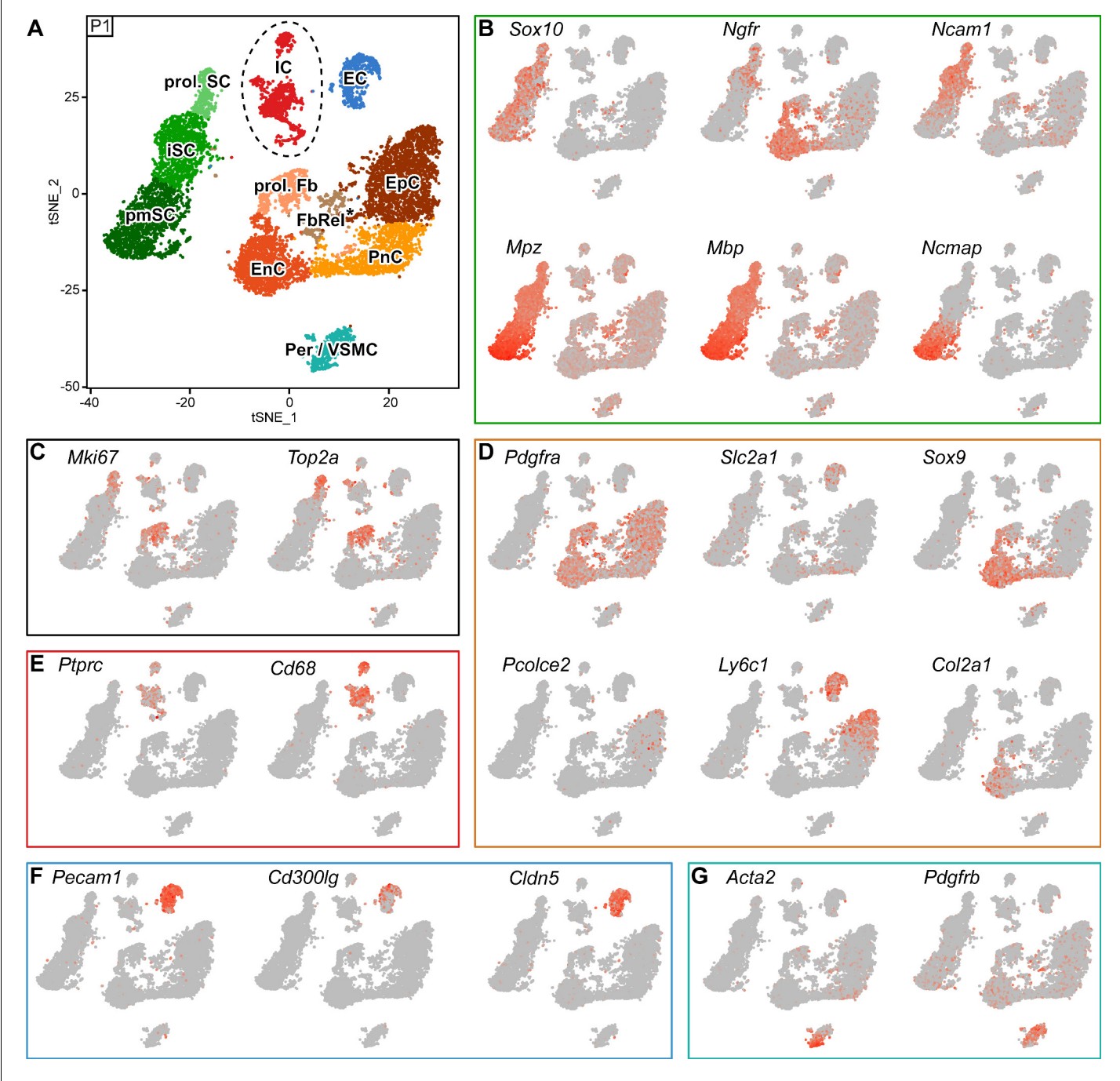

**Figure 6.** RNA single-cell profiling of sciatic nerve cells handled with 10x Genomics at postnatal day 1. (**A**) tSNE plot depicting the major clusters of nerve resident cells at P1 and handled with 10x Genomics. These include single clusters of pericytes and vascular smooth muscle cells (Per/VSMC), endothelial cells (EC), immune cells (IC), perineurial cells (PnC), endoneurial cells (EnC), and epineurial cells (EpC). Schwann cells (SCs) are represented as three clusters, including proliferating Schwann cells (prol. SC), immature Schwann cells (iSC) and pro-myelinating Schwann cells (pmSCs). In addition, another cluster expressing markers typical of proliferating cells was marked as proliferating fibroblast-like cells (prol. Fb), and a cluster located between EnC, PnC and EpC was marked as fibroblast-related cluster (FbRel*). The tentative nature of this cluster is indicated by an asterisk (see text). Dashed lines indicate a merge of clusters produced from the default output of the analysis (see *Figure 6—figure supplement 2*). (**B–G**) tSNE plots overlaid with the detected relative transcript levels. A gradient of red tones indicates relative abundance of each transcript per cell, in which dark red indicates higher levels, and gray indicates not detected. The markers described here label the indicated cluster also at P60 (see *Figure 5*), although not always exclusively. (**B**) All SC clusters are labeled by expression of *Sox10*. The iSC and prol. SC clusters are labeled by *Ncam1* and *Ngfr*, whereas the pmSC cluster is labeled by *Ncmap* and is enriched in *Mpz* and *Mbp*. (**C**) The proliferating cell clusters (prol. SC and prol. Fb) are labeled by *Mki67* and *Top2a*.

*Figure 6 continued on next page*

*Figure 6 continued*

(**D**) tSNE plots overlaid with the detected relative transcript levels for EnC, PnC, and EpC clusters. All three clusters are labeled by *Pdgfra*. The PnC cluster is moderately labeled by *Slc2a1*, the EnC cluster is labeled by *Sox9* and *Col2a1*, and the EpC cluster is labeled by *Ly6c1* and *Pcolce2*. (**E**) The IC cluster is labeled by *Ptprc* and *Cd68*. (**F**) The EC cluster is labeled by *Pecam1*, *Cldn5*, and in part *Cd300lg*. (**G**) The Per/VSMC cluster is labeled by *Acta2* and *Pdgfrb*. See also *Figure 6—figure supplement 1* and *Figure 6—figure supplement 2*.

The online version of this article includes the following figure supplement(s) for figure 6:

**Figure supplement 1.** Integrated tSNE plot of nerve cells handled with 10x Genomics at P1, encoding the chromium run that each cell derives from, the most enriched transcripts per cluster, and the gene detection depth per cell.

**Figure supplement 2.** Original clusters overlaid with tSNE plots related to single-cell sequencing analysis of sciatic nerve cells handled with 10x Genomics at P1.

stage (*Figure 6F*). Finally, a pericyte and vascular smooth muscle cell cluster could be identified by expression of *Acta2* and *Pdgfrb* (marked Per/VSMC; *Figure 6G*). Row z-scores of the ten genes with the most enriched transcripts per cluster within the dataset are included as a heat map projection (*Figure 6—figure supplement 1B*), and a more comprehensive list is also provided (*Supplementary file 3*). Further technical details concerning the sequencing data, overlaid with individual cells of the tSNE projection, include the number of detected genes (*Figure 6—figure supplement 1C*), the fraction of mitochondrial content (*Figure 6—figure supplement 1D*), and the number of unique molecular identifiers (*Figure 6—figure supplement 1E*).

In summary, single-cell RNA sequencing of all-included sciatic nerves with 10x genomics provided transcriptome profiles on most of the major cell types that are expected to reside in mouse sciatic nerves, both in early postnatal development and in young adults. One notable exception are epineurial adipocytes (*Montani et al., 2018*), which we did not detect as a separate cluster. A few cells expressing adipocyte markers, including *Car3* (encoding Carbonic Anhydrase 3), *Cdo1* (encoding Cysteine Dioxygenase Type 1), *Adipoq* (encoding Adiponectin), and *Plin1* (encoding Perilipin 1) (*Burl et al., 2018*; *Deng et al., 2015*), can be found within the P60 epineurial cells cluster (see SNAT), but it remains unclear at this time whether they represent bona fide epineurial adipocytes. Another peculiar example are cells that line potentially lymphatic vessels. A small group of cells with expression of markers of lymphatic endothelial cells including *Prox1* (encoding the transcription factor Prox-1), *Flt4* (encoding VEGF Receptor-3), and *Lyve1* (encoding Lymphatic Vessel Endothelial Hyaluronan Receptor 1) (*Wigle et al., 2002*) can be found within one of the P60 endothelial cell cluster (endothelial cell cluster marked EC2; see SNAT), but they have not been allocated as separate cluster potentially due to their scarcity.

We note that the approach of using dissociation of cells from nerves followed by FACS has the limitation of some potential subsampling of a subset of cells (*Figure 2*). In particular, SCs isolated from extensively myelinated nerves might not reveal the full diversity spectrum. The observed low relative extraction efficiency in this context may reflect different sensitivities of distinct SC populations to the experimental procedures, associated with the potential of subsampling.

On an outlook note, our data can now be used for detailed comparative mapping of expression of particular genes, at the cellular nerve level, as the fundamental basis to understand peripheral nerve biology in health and disease. As some applied examples, we recommend to search SNAT for your favorite individual gene, or gene family, or genes that encode components of the various functional structures established by the different nerve-resident cell types, or 'orphan' genes with incompletely understood functions.

Taken together, the validated datasets presented in this study are intended to serve as a reference point of gene expression in peripheral nerves, obtained by various complementary forms of RNA sequencing techniques. Our multilayered approach allowed both the coverage and emphasis on SCs and other major populations of resident cells in sciatic nerves, mainly focused on postnatal development. We have integrated our data in a freely accessible and searchable web interface (https://www.snat.ethz.ch) with the intention to make a contribution to the filling of a gap in current peripheral nerve research. Using this resource as baseline, together with recently published complementary extensions mainly focusing on immune cells in peripheral nerves (*Kalinski et al., 2020*; *Wang et al., 2020*; *Ydens et al., 2020*) and adult nerves (*Wolbert et al., 2020*), we anticipate that there will be expanding efforts to cover various aspects of peripheral nerve biology such as in cellular-, injury-, aging-, and disease models (*Toma et al., 2020*) as well as covering other related glia

cells (*Avraham et al., 2020*; *Stierli et al., 2019*), hopefully yielding insightful and valuable data as the fundamental basis of building and testing hypothesis in functional follow-up studies.

# Materials and methods

## Key resources table

| Reagent type (species) or resource | Designation | Source or reference | Identifiers | Additional information |
|---|---|---|---|---|
| Genetic reagent (*M. musculus*) | GtROSA26Sor < tm1(EYFP)Cos> | Jackson Laboratory | RRID:IMSR_JAX:006148 | PMID:11299042 |
| Genetic reagent (*M. musculus*) | B6.Cg-Tg(CAG-Ds Red*MST)1Nagy/J | Jackson Laboratory | RRID:IMSR_JAX:006051 | PMID:15593332 |
| Genetic reagent (*M. musculus*) | Tg(MPZ-cre)26Mes/J | Jackson Laboratory | RRID:IMSR_JAX:017927 | PMID:10586237 |
| sequence-based reagent | Genotyping primer: DsRed forward | Jackson Laboratory Protocol 22460 Standard PCR Assay - Tg(DsRed), version 1.3 | | 5'-CCCCGTAATGCAGAAGAAGA-3' |
| Sequence-based reagent | Genotyping primer: DsRed reverse | Jackson Laboratory Protocol 22460 Standard PCR Assay - Tg(DsRed), version 1.3 | | 5'-GGTGATGTCCAGCTTGGAGT-3' |
| Sequence-based reagent | Genotyping primer: Cre forward | PMID:30648534 | | 5'-ATCGCCAGGCGTTTTCTGAGCATAC-3' |
| Sequence-based reagent | Genotyping primer: Cre reverse | PMID:30648534 | | 5'-GCCAGATTACGTATATCCTGGCAGC-3' |
| Sequence-based reagent | Genotyping primer: Rosa26-loxPstoploxP -YFP forward | PMID:15905512 | | 5'-AAAGTCGCTCTGAGTTGTTAT-3' |
| Sequence-based reagent | Genotyping primer: Rosa26-loxPstoploxP-YFP wildtype reverse | PMID:15905512 | | 5'-GGAGCGGGAGAAATGGATATG-3' |
| Sequence-based reagent | Genotyping primer: Rosa26-loxPstoploxP-YFP transgenic reverse | PMID:15905512 | | 5'-GCGAAGAGTTTGTCCTCAACC-3' |
| Sequence-based reagent | Oligo-dT30VN | PMID:24385147 | | 5'-Biot–AAGCAGTGGTATCAACGC AGAGTACT30VN-3' |
| Sequence-based reagent | Template-switch oligo (TSO) | PMID:24385147 | | 5'-Biot-AAGCAGTGGTATCAACGCA GAGTACATrGrG +G-3'(rG = riboguanosine; +G = LNA-modified guanosine) |
| Sequence-based reagent | ISPCR | PMID:24385147 | | 5'-Biot-AAGCAGTGGTATCAACGCAGAGT-3' |
| Sequence-based reagent | Nextera XT 384 UDIs primer set | Integrated DNA Technologies | | |
| Commercial assay or kit | TruSeq Stranded mRNA Sample Prep Kit | Illumina, San Diego, CA, USA | 20020595 | |
| Commercial assay or kit | SuperScript II Reverse Transcriptase | Thermo Fischer | 18064014 | |
| Commercial assay or kit | KAPA HiFi HotStart ReadyMix | Roche | KK2601 | |
| Commercial assay or kit | Illumina Nextera XT Index Kit | Illumina, San Diego, CA, USA | FC-131–1096 | |
| Commercial assay or kit | Chromium Single Cell 3' Library and Gel Bead Kit v2 | 10X Genomics | PN-120237 | |
| Commercial assay or kit | Chromium Single Cell A Chip Kit | 10X Genomics | PN-120236 | |

*Continued on next page*

*Continued*

| Reagent type (species) or resource | Designation | Source or reference | Identifiers | Additional information |
|---|---|---|---|---|
| Commercial assay or kit | Chromium i7 Multiplex Kit | 10X Genomics | PN-120262 | |
| Commercial assay or kit | Trypsin | Sigma Aldrich, St. Louis, MO, USA | Cat#T9201 | |
| Commercial assay or kit | Collagenase type 2 | Worthington Biochemical Corporation, NJ, USA | Cat#LS004174 | |
| Commercial assay or kit | Hyaluronidase | Worthington Biochemical Corporation, NJ, USA | Cat#LS005474 | |
| Software, algorithm | Illustrator version CC | Adobe | | |
| Software, algorithm | STAR Aligner (v2.5.4b; v2.6.1.c) | PMID:23104886 | RRID:SCR_015899 | |
| Software, algorithm | Rsubread (v1.32.4) | PMID:23558742 | RRID:SCR_016945 | |
| Software, algorithm | R package Seurat (V2.3.1) | PMID:29608179; PMID:25867923 | RRID:SCR_016341 | |
| Software, algorithm | Slingshot (v1.0.0) | PMID:29914354 | RRID:SCR_017012 | |
| Software, algorithm | CellRanger (v2.0.2) | 10x Genomics | RRID:SCR_017344 | |
| Software, algorithm | fastICA package (1.2–2) | PMID:10946390 | RRID:SCR_013110 | |

## Experimental animals

The peripheral nerves used for the timeline shown in *Figure 1* and respective supplement were extracted from C57bl6/J wild-type animals at multiple ages, as indicated in the text and figure legends. Experiments which required sorting of cells by FACS (*Figures 2–6*, including respective supplements) were performed with cells acutely extracted from mice expressing MPZCre (Tg(Mpz-cre)26Mes/J; RRID:IMSR_JAX:017927) (designated in this manuscript as P0Cre) (*Feltri et al., 1999*), which promotes the recombination of the reporter gene Rosa26-loxPstoploxP-YFP mouse line (GtROSA26Sor < tm1(EYFP)Cos>; RRID:IMSR_JAX:006148) (*Srinivas et al., 2001*) (designated in this manuscript as ReYFP^Tg) to specifically label SCs with expression of enhanced yellow fluorescent protein (YFP). Furthermore, these mice also expressed DsRed.MST reporter protein under the control of the chicken beta actin promoter coupled with the cytomegalovirus (CMV) immediate early enhancer (B6.Cg-Tg(CAG-DsRed*MST)1Nagy/J; RRID:IMSR_JAX:006051) (*Vintersten et al., 2004*) (designated in this manuscript as βAct-DsRed^Tg). The combination of these reporter proteins was used to select SCs and other nerve-resident cells as described in the figure legends and text. Note that all mice used in this study were not perfused prior to tissue extraction to minimize handling time and tissue alterations.

Mice of both sexes were used in the study, as indicated in the respective figure legends. All mice used are on the C57bl/6J background. Mice were allocated to experiments based on their age and genotype, but otherwise in a random manner. Mice were housed in a maximum of 5 animals/cage, kept in a 12 hr dark-light cycle, and provided ad libitum access to standard chow and water. No computation methods were used to estimate sample size, and these were selected according to sample sizes generally employed in the research field. Genotypes were determined by genomic PCR using the following primer sets: Cre forward 5'-ATCGCCAGGCGTTTTCTGAGCATAC-3', reverse 5'-GCCAGATTACGTATATCCTGGCAGC-3'; Rosa26-loxPstoploxP-YFP forward 5'-AAAGTCGCTCTGAGTTGTTAT-3', transgenic reverse 5'-GCGAAGAGTTTGTCCTCAACC-3', wildtype reverse 5'-GGAGCGGGAGAAATGGATATG-3'; DsRed forward 5'-CCCCGTAATGCAGAAGAAGA-3', reverse 5'-GGTGATGTCCAGCTTGGAGT-3'. All animal experiments were performed with the approval and in accordance to the guidelines of the Zurich Cantonal Veterinary Office under permit ZH090/2017.

## Preparation of cell suspension and FACS

Sciatic nerves were extracted from P0Cre[-/+]: ReYFP[Tg/+]: βAct-DsRed[Tg/+] mice and collected in ice-cold PBS. The protocol to extract cells from these nerves was adapted from *Byrne, 2016*. Nerves were transferred onto a 6 cm cell-culture dish and minced using a scalpel. Minced nerves were transferred to a 15 ml centrifugation tube and were enzymatically dissociated on a thermoshaker (Eppendorf ThermoMixer C) at 37°C for 20 min with a solution containing 0.15% Trypsin (Sigma, #T9201), 0.3% collagenase Type 2 (Bioconcept, #LS004174), 0.04% hyaluronidase (Bioconcept, #LS005474) in HBSS. Ten minutes after the start of enzymatic digestion, the tissue was mechanically dissociated by pipetting using a 1 ml precision pipette. At the end of the 20 min digestion, the tissue was completely mechanically dissociated by pipetting 20 times with a 200 µl precision pipette, and the enzymatic reaction was stopped by adding 10% FCS in PBS. The cell suspension was centrifuged (Eppendorf Centrifuge 5810 R) at 300 g for 5 min at 4°C, and the resulting pellet was resuspended in FACS buffer (0.5% BSA, 5 mM EGTA, in PBS). Before cell sorting, the mix was filtered through a 50 µm filter.

Cells were sorted using a SH800S Sony cell sorter. For P5 bulk RNA sequencing, SCs were sorted using eYFP intensity and SSC-A into 1.5 ml Eppendorf tubes (*Figure 2A*). For single-cell sequencing of Schwann cells, eYFP-positive cells were sorted onto 384-well plates followed by the Smart-seq2 protocol. For single-cell sequencing of whole sciatic nerve lysates, DsRed-positive cells were sorted into 1.5 ml Eppendorf followed by the 10x genomics protocol. At P1, a negative selection of DsRed-eYFP-double-positive cells additionally was applied to restrict the SC abundance to approximately 30% of the sorted cells processed with the 10x genomics protocol.

## Bulk RNA sequencing of sciatic nerve lysates

Immediately after dissection, sciatic nerves were placed in ice-cold PBS, stripped from epineurium and perineurium, snap-frozen in liquid nitrogen and stored at −80°C. On the day of RNA extraction, the nerves were grinded while frozen into a fine frozen powder. Subsequently, total RNA was extracted using Qiazol (Qiagen) as per manufacturer's instructions. Using RNA extracted from peripheral nerves, the libraries were prepared following the Illumina TruSeq stranded mRNA protocol. The quality of the RNA and final libraries was determined using an Agilent 4200 TapeStation System. The libraries were pooled equimolecularly and sequenced in an Illumina HiSeq4000 sequencer (single-end 125 bp) with a theoretical depth of approximately 20 Mio reads per sample.

## Bulk RNA sequencing of Schwann cells

Using RNA from sorted SCs, the libraries were prepared following the Smart-seq2 protocol (*Picelli et al., 2013*; *Picelli et al., 2014*). The quality of the RNA, cDNA and final libraries was determined using an Agilent 4200 TapeStation System. 0.5 ng of cDNA from each sample were tagmented and amplified using Illumina Nextera XT kit. The resulting libraries were double-sided size selected (0.5x followed by 0.8x ratio using Beckman Ampure XP beads), quantified and pooled equimolecularly. The pool of libraries was sequenced in an Illumina HiSeq4000 sequencer (single-end 125 bp) with a depth of approximately 20 Mio reads per sample.

## Single-cell RNA sequencing of Schwann cells

The transcriptome of the single SCs was analyzed using a miniaturized version of the Smart-seq2 protocol with the help of a SPT Labtech Mosquito HV pipetting robot (*Jaeger et al., 2020*). Briefly, single cells were directly sorted into 384-well plates containing 0.8 µl of lysis buffer (0.1% vol/vol Triton X-100, 2.5 mM dNTPs, 2.5 µM oligo-dT, 1 U/µl Promega RNasin Plus RNase inhibitor). Reverse transcription was performed in a final volume of 2 µl followed by cDNA amplification in a final volume of 5 µl. The quality of the cDNAs was evaluated using an Agilent 2100 Bioanalyzer. 0,1 ng of cDNA from each cell on the plate was individually tagmented using Illumina Nextera XT kit in a final volume of 5 ul, followed by barcoding and library amplification in a final volume of 10 ul. The resulting 384 libraries were pooled, double-sided size selected (0.5x followed by 0.8x ratio using Beckman Ampure XP beads) and quantified using an Agilent 4200 TapeStation System. The pool of libraries was sequenced in an Illumina HiSeq4000 with a depth of approximately 500,000 reads per cell (around 200 Mio reads per plate).

### Single-cell RNA sequencing of sciatic nerve cells

Whole nerve cells suspensions were sorted and the quality and concentration of the single-cell preparations were evaluated using an haemocytometer in a Leica DM IL LED microscope and adjusted to 1000 cells/μl. A total of 4000 cells per sample were loaded into the 10x Genomics Chromium controller and library preparation was performed according to the manufacturer's indications (single cell 3′ using v2 chemistry). The resulting libraries were sequenced in an Illumina HiSeq2500 sequencer according to 10x Genomics recommendations (paired-end reads, R1 = 28, i7 = 8, R2 = 91) to a depth of approximately 50,000 reads per cell.

### Data analysis of sciatic nerve bulk RNA sequencing and Schwann cell bulk RNA sequencing

Adapters and low-quality tails were trimmed from reads before mapping to transcriptome. STAR aligner (v2.6.1.c) (*Dobin et al., 2013*) was used to align the RNA-seq data to Ensembl release 91 reference genome build GRCh38.p10. Gene expression values were quantified by featureCounts from the Bioconductor package Rsubread (v1.32.4) (*Liao et al., 2013*).

### Single-cell RNA sequencing data analysis and integration of Schwann cells handled with Smart-Seq2 protocol

All reads from Smart-Seq2 plates were mapped to Ensemble release 91 reference genome build GRCh38.p10 with STAR aligner (v2.5.4b) (*Dobin et al., 2013*) and counted by *featureCounts()* from Rsubread package (v1.32.4) (*Liao et al., 2019*). The downstream analysis was performed with R package Seurat (v2.3.1) (*Butler et al., 2018*; *Satija et al., 2015*) for each dataset in different parameters. Cells with fewer than 1500 detected genes were discarded. Library-size normalization was done for each cell by scaling by the total number of transcripts and multiplying by 100,000. The data was then natural-log transformed for all downstream analyses. The detection of highly variable genes was based on the average expression and dispersion for each gene using the default parameters (x. low.cutoff = 0.1, x.high.cutoff = 8, y.cutoff = 1) of the function FindVariableGenes() in Seurat. The dimensionality of the data set was reduced using principal component analysis (PCA), with 7 and 20 principal components selected for each time point and integrated dataset, respectively. Unsupervised clustering of the cells was performed with the shared nearest neighbor (SNN) clustering algorithm implemented in the FindClusters() function in Seurat with a resolution of 0.4 for P1 and P5, 0.3 for P14 and P60, 0.6 for the integrated dataset. tSNE plots were generated using the same PCs that were input to the clustering analysis and setting the perplexity parameter to 30. The two plates from P60 were batch corrected and integrated by Seurat's Canonical Correlation Analysis implementation with 12 canonical vectors. A resolution of 0.5 was used in the clustering step. Differentially expressed genes (cluster markers) for each of the cell populations were determined using Wilcoxon rank sum test. We used the program Slingshot (v1.0.0) (*Street et al., 2018*) to infer cell lineages. As an input to Slingshot, we used a matrix representing the cells in a reduced-dimensional space and a vector of cluster labels, as inferred by the above analyses with Seurat. The starting cluster was set to 'proliferating Schwann cells'. Slingshot identified two branching trajectories, one corresponding to myelinating Schwann cells, another corresponding to non-myelinating (Remak) Schwann cells. Then, pseudotimes on each lineage were estimated with the implementation of fastICA() in package fastICA (*Hyvärinen and Oja, 2000*) from single-cell gene expression data. To visualize the gene expression change over pseudotime, we fitted a generalized additive model (GAM) on the pseudotime with smoothing spline.

### Single-cell RNA sequencing data analysis of nerve cells handled with 10x Genomics

CellRanger (v2.0.2) from 10x Genomics was used to demultiplex, align the reads to Ensembl reference build GRCh38.p10 and collapse UMIs. Starting from the filtered gene-cell count matrix by CellRanger's default cell calling algorithm, we applied the Seurat workflow, as described in Smart-Seq2 data analysis section. Cells from the three independent runs were pooled together. To exclude low-quality cells or possible doublets, we filtered out cells for which fewer than 500 genes or more than 5000 genes were detected, leaving 11339 cells at P1 and 12428 cells at P60 for further analyses. FindVariableGenes() function from Seurat identifies 2145 and 2240 highly variable genes at P1 and

P60 respectively, with parameters (x.low.cutoff = 0.0125, x.high.cutoff = 3, y.cutoff = 0.5, dispersion. function = LogVMR). In the dimensionality reduction step, 10 principal components were selected. A resolution of 0.5 was chosen in the unsupervised clustering algorithm at P1, while 0.4 was used at P60.

The clusters identified with 10x Genomics at both P1 and P60 were refined by merge and division of some adjacent clusters, based on the overarching cell type they represent. In *Figure 5—figure supplement 3* and *Figure 6—figure supplement 2* we included the tSNE plots overlaid with the default clusters derived from the clustering analysis at both time points. At P60 (*Figure 5—figure supplement 3*), clusters 4 and 10 were merged in a single cluster labeled (IC) because both clusters represent typical markers of immune cells. Furthermore, we merged clusters 2 and 9, since both represent markers typical of perineurial cells (PnC). Clusters 1, 0, and 12 were also merged, since these cells are enriched in markers characteristic of epineurial cells (EpC). Cluster 7 was originally composed of two islands which were separated. The reasoning is explained in the *results and discussion* section. Since the lower island of cluster seven expresses both *Pdgfrb* and *Pecam1*, we cannot confidently state these are bona fide pericytes. This island was therefore re-labeled as cluster (Per/EC*). The remaining island in cluster 7 is more enriched in *Pdgfrb* and likely represents pericytes, and cluster 8 is enriched in *Acta2* and likely represents vascular smooth muscle cells. The merged cluster adopts the name of both cell types, as pericyte and vascular smooth muscle cell cluster (Per/VSMC). At P1 (*Figure 6—figure supplement 2*), we merged the cluster 5 and 11 into one cluster labeled (IC), since both original clusters represent typical markers of immune cells.

## Preparation of figure panels

Figure panels were assembled with Adobe Illustrator (version CC).

## Data and code availability

RNA-sequencing data have been deposited in the GEO database under accession number GSE137870. The SNAT website can be reached at https://www.snat.ethz.ch. The Seurat objects used to produce the main figures in this publication are available at https://www.snat.ethz.ch/seurat-objects.html. The code to process, analyze and visualize all the sequencing data is available at https://github.com/suterlab/SNAT-code-used-for-primary-data-analysis copy archived at [swh:1:rev:3a94c4740a9c40ece69ce7a620c15757e61d0628]; *Gerber, 2021*.

## Acknowledgements

We thank all members of the Suter lab for discussions, the Functional Genomics Center Zürich for excellent technical support, Drs. Maria Laura Feltri and Lawrence Wrabetz for transgenic mice, and Drs. Igor Adameyko and Maria Eleni Kastriti for valuable support.

## Additional information

### Funding

| Funder | Author |
| --- | --- |
| Eidgenössische Technische Hochschule Zürich | Ueli Suter |

The funders had no role in study design, data collection and interpretation, or the decision to submit the work for publication.

### Author contributions

Daniel Gerber, Conceptualization, Data curation, Formal analysis, Validation, Investigation, Visualization, Methodology, Writing - original draft, Writing - review and editing; Jorge A Pereira, Conceptualization, Data curation, Formal analysis, Validation, Investigation, Visualization, Writing - original draft, Writing - review and editing; Joanne Gerber, Conceptualization, Data curation, Formal analysis, Investigation, Visualization, Methodology, Writing - review and editing; Ge Tan, Data curation, Formal analysis, Validation, Investigation, Methodology, Writing - original draft, Writing - review and

editing; Slavica Dimitrieva, Data curation, Formal analysis, Investigation, Methodology, Writing - review and editing; Emilio Yángüez, Formal analysis, Investigation, Methodology, Writing - original draft, Writing - review and editing; Ueli Suter, Conceptualization, Resources, Data curation, Formal analysis, Supervision, Funding acquisition, Project administration, Writing - review and editing

### Author ORCIDs
Daniel Gerber http://orcid.org/0000-0001-6613-654X
Jorge A Pereira https://orcid.org/0000-0002-0159-4133
Ge Tan http://orcid.org/0000-0003-0026-8739
Ueli Suter https://orcid.org/0000-0002-9211-5184

### Ethics
Animal experimentation: All animal experiments were performed with the approval and in accordance to the guidelines of the Zurich Cantonal Veterinary Office under permit ZH090/2017.

### Decision letter and Author response
Decision letter https://doi.org/10.7554/eLife.58591.sa1
Author response https://doi.org/10.7554/eLife.58591.sa2

## Additional files
### Supplementary files
• Supplementary file 1. Detected RPKM values in bulk RNA sequencing datasets. The file depicts the RPKM values for each sample submitted to bulk RNA sequencing derived from peripheral nerve tissue (related to *Figure 1*) and from sorted Schwann cells (related to *Figure 2*). Each dataset is displayed in individual tabs.

• Supplementary file 2. Identity of individual cells in the single-cell RNA sequencing datasets. The assigned cell type of each cell from the merged Smart-Seq2 dataset (related to *Figure 4*), the 10x P60 runs (related to *Figure 5*), and the 10x P1 runs (related to *Figure 6*), together with the cell embedding on tSNE plots. Each dataset is displayed in individual tabs.

• Supplementary file 3. Relative enrichment of individual genes per single-cell sequencing cluster. The file depicts the positive markers of each cell type (cluster) with the average marker gene expression in each cell type (cluster) from the merged Smart-Seq2 samples (related to *Figure 4*), the 10x P60 runs (related to *Figure 5*), and the 10x P1 runs (related to *Figure 6*), each displayed in individual tabs. The significance level (p_val), fold change (avg_logFC), and proportion of cells in the cluster that express the gene (pct.1 and pct.2) are also included. The positive marker identification was done by *FindAllMarkers()* from Seurat. For each cluster, genes with a p-value below 0.05 are considered differentially enriched.

• Transparent reporting form

### Data availability
RNA-sequencing data have been deposited in the GEO database under accession number GSE137870. The SNAT website can be reached at https://www.snat.ethz.ch. The Seurat objects used to produce the main figures in this publication are available at https://www.snat.ethz.ch/seurat-objects.html. The code to process, analyze and visualize all the sequencing data is available at https://github.com/suterlab/SNAT-code-used-for-primary-data-analysis copy archived at https://archive.softwareheritage.org/swh:1:rev:3a94c4740a9c40ece69ce7a620c15757e61d0628. All other data generated or analysed during this study are included in the manuscript and supporting files.

The following dataset was generated:

| Author(s) | Year | Dataset title | Dataset URL | Database and Identifier |
|---|---|---|---|---|
| Gerber D, Pereira | 2019 | The transcriptome of sciatic | https://www.ncbi.nlm. | NCBI Gene Expression |

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
