## [Decision Letter]

**Acceptance summary:**

Gerber et al. provide a rich data resource of mouse sciatic nerve transcriptional profiles across a range of ages and cellular resolutions. The authors performed both bulk and single cell RNA sequencing, documenting gene expression changes among the diverse cell types found in this peripheral nerve, and include specific controls to ensure the quality of the resource. Importantly, they also developed a web-based interface to provide the community ready access to these data. This resource will be of significant value to the peripheral nerve community, as a platform to understand the molecular basis of sciatic nerve development and function.

**Decision letter after peer review:**

Thank you for submitting your article "Transcriptional Profiling of Mouse Peripheral Nerves to the Single-Cell Level to Build a Sciatic Nerve ATlas (SNAT)" for consideration by *eLife*. Your article has been reviewed by 3 peer reviewers, and the evaluation has been overseen by a Reviewing Editor and Marianne Bronner as the Senior Editor. The following individuals involved in review of your submission have agreed to reveal their identity: Loyal A Goff (Reviewer #2); Ahmet Hoke (Reviewer #3).

The reviewers have discussed the reviews with one another and the Reviewing Editor has drafted this decision to help you prepare a revised submission.

Summary:

This study provides a rich resource of transcriptional profiles for the mouse sciatic nerve across a range of temporal dynamics and cellular resolutions. The authors begin with a well-designed bulk RNA-Seq study of the dissected mouse sciatic nerve across developmental and postnatal time points. This is followed by a bulk RNA-Seq study of sorted schwann cells (SC) in a select subset of time points to identify the transcriptional signatures of the enriched SC component(s) of the sciatic nerve, isolating both myelinating and unmyelinating subtypes. In total, this effort provides a more focused transcriptional survey of SC cell subtypes/states. Further analysis is done on the FACS sorted SC cells at single cell resolution and finally, the authors perform a comprehensive single cell analysis of the entire sciatic nerve, including non-SC cells. The authors do a commendable job at presenting the raw data and describing the specific quality controls required to assure the quality of the resource. However, there are a few analysis choices and mis-interpretations of differential gene expression that raise significant concern for acceptance of the paper in its current form. There is little to no novel biology uncovered or described in this resource paper, albeit that the intent behind this manuscript is to present the data, not necessarily 'discover' anything from it. Where the authors do attempt to describe potentially novel findings, these are not directly supported by the data alone and it should be noted that these conclusions are speculative requiring validation. Furthermore, while the comprehensiveness of this manuscript in terms of the data generated is impressive, the lack of integrative analysis across datasets, and the relative 'light-touch' analysis for each component of this dataset, is a significant shortfall, particularly as several similar datasets have recently been published.

Essential revisions:

1. The authors describe a publicly available searchable web interface (www.snat.ethz.ch). However, access to this resource is currently password-protected. This website needs to be made available to the reviewers in order to evaluate the utility and stability of this public resource.

2. Many of the axes are unlabeled (e.g. Figures 3 -6…). Please add labels to the figures directly.

3. The clusters presented in Figure 3 Panel G (P60) are learned on a reduced dimensional embedding that the authors understand to be biased with respect to batch. Indeed, this is a particular focus of Figure 3—figure supplement 1 panel D. It is commendable that the authors note this and correct with batch correction in the supplement, however the clusters used to annotate mSC2 and mSC3 are derived from the original, biased embedding. These clusters disproportionately contain cells from each batch and therefore the differential genes between mSC2 and mSC3 in Figure 3—figure supplement 1 panel A (P60) primarily represent genes that differ between batches. This should be addressed prior to acceptance.

Given the concerns about batch:cluster confounding above, the designation of mSC2 and mSC3 in the joint embedding remains concerning as well. Potentially confounding the interpretation of pseudotime in Figure 4 and Figure 4—figure supplement 1. Many of the genes identified along pseudotime for mSC3 overlap with the mSC3-specific genes from Figure 3—figure supplement 1 panel A (P60).

4. Figure 4: All axes are unlabelled, including the 'pseudotime values' along X and the gene expression levels along Y for Panel D. It is unclear that the slingshot pseudotime trajectories are resolving the best-fit paths through the data instead of potentially being biased by arbitrary cluster assignments. For example, the designation of the tSC cluster as a discrete and/or transient state is not clear from the embedding. However, the designation of this as a cluster will force slingshot to produce a trajectory through the tSC cluster when this may not be borne out directly from the data. If the authors wish to support these trajectories as appropriate paths towards specification, it is highly recommended that they further perform an RNAVelocity analysis on the joint, embedded single cell analysis in Figure 4. This will provide an 'annotation-agnostic' representation of the change in predicted state over the chosen embedding. The hope is that the velocity field mirrors the paths identified by slingshot. This will provide compelling evidence that the trajectories presented are correct and not 'winding' unnecessarily.

There are some important details missing from the slingshot methods and analysis. The authors use a GAM model to identify genes with significant differential expression WRT pseudotime, but the details of the model are not presented, and the choice of significance thresholds is not described. Further, the authors do not present a list of genes identified by the GAM model, instead relying on Z-scores for previous cluster assignment specificity, (Figure 4—figure supplement 1 panel B) instead of the significance values from the pseudotime fitting. This does not use the pseudotime analysis appropriately but rather still relies on the 'aggregate' expression of genes across clusters.

In general, the application of pseudotime is not complete, and the authors do not use this analysis approach to appropriately resolve trajectories and gene expression differences along these trajectories. The analysis presented is still restricted to 'cluster-level' aggregate differences.

The authors interpret the "appearance of multiple mSC clusters as a peculiar feature in the tSNE plots". This is true and due, in part, to both the nature of tSNE (which does not by default preserve global structure in embeddings) and the previously noted concerns about batch (in P60) that are not addressed in the joint embedding. The authors however, choose to interpret the different mSC clusters as biologically motivated. (lines 361-368). This statement should be re-evaluated and our couched.

A semantic point, the authors claim three 'biological' replicates for the 10x phase of their study per time point. Since each replicate consists of two nerves (one from male and one from female) these are not technically biological replicates but rather technical replicates.

5. Figure 5: It is concerning that for the 10x study, SCs account for only 3.3% of the identified cell types, and yet in the bulk analysis presented in the beginning, only the SC marker genes and how they change across time points are presented. Are the SC gene signatures truly dominating in the bulk samples presented, and as suggested by the authors for the motivation for this study? This is further evidenced by the diagram provided in Figure 5—figure supplement which suggests that the vast majority of cells in this dataset should be Schwann cells. It would be very useful to know, or at least retrospectively estimate, what proportion of the gene expression signatures in the bulk study is derived from SC cells or cells of other cell types. Learning this information from the single cell data and applying this type of analysis to their bulk data would further strengthen the application and utility of this resource and demonstrate a clear need to re-evaluate the use of bulk studies in the PNS.

6. Figure 6: The authors 'identify clusters of proliferating fibroblasts labelled by Pdgfra expression' and reason that these clusters represent biologically interesting/unique clusters of endoneurial fibroblast-lik cells, perineuial cells, or epineurial cells, and represent a 'mysterious cluster' It is highly probable that the expression of Pdgfra in these clusters is not biological, but rather a consequence of either A) incomplete dissociation from true Pdgfra-expressing cells, or B) background signal detected in otherwise low-quality cells, (both appropriately indicated by the authors). The former is supported by the higher-than average number of genes detected in these populations (consistent with potential doublets), despite the authors use of a 'doublet detection' algorithm. The latter is supported by the higher mitochondrial read ratios observed for these clusters as well. Without supporting in situ hybridizations to support the identification and co-expression of Pdgfra in these populations, this finding should be removed or addressed in significantly greater detail. The authors claim(s) that "we cannot exclude that some cells in this cluster might also be in a hypothetical transitory stage of neural crest-derived cells, potentially expressing markers of multiple lineages on their route to differentiation" is a purely artificial explanation that is not supported by the evidence provided. Suggestion of this as an alternative explanation does not satisfy an Occom's razor explanation and must be substantiated with further experimentation.

The authors state that " Clusters representing the different cell types were overall in much closer proximity to each other on the tSNE plots at P1 and P60" and use this to suggest that this may represent "potential cell lineage transitions or cell plasticity at early stages…". This is not an appropriate or valid conclusion to be drawn from tSNE plots. The meaningful proximity of clusters in a tSNE embedding cannot be directly inferred and is a common mis-interpretation of the method used. Please see discussion here for cautions and guidelines for interpretation of tSNE embeddings (https://mlexplained.com/2018/09/14/paper-dissected-visualizing-data-using-t-sne-explained).

7. Given the potential utility of this dataset as a resource for the PNS community, it is strongly recommended that the authors make their analysis (scripts/code/etc) publicly available for review.

[Editors' note: further revisions were suggested prior to acceptance, as described below.]

Thank you for submitting your article "Transcriptional Profiling of Mouse Peripheral Nerves to the Single-Cell Level to Build a Sciatic Nerve ATlas (SNAT)" for consideration by *eLife*. Your article has been reviewed by 3 peer reviewers, and the evaluation has been overseen by a Reviewing Editor and Marianne Bronner as the Senior Editor. The following individuals involved in review of your submission have agreed to reveal their identity: Roman J Giger (Reviewer #1); Loyal A Goff (Reviewer #2); Ahmet Hoke (Reviewer #3).

Essential revisions:

1. Please address concerns about the validity of conclusions related to the pseudotime analysis raised by Reviewer #2. If additional analysis is possible, please include, otherwise modify the text to temper conclusions that can be made using this approach.

2. Please add additional discussion of potential bias that may arise from subsampling a subset of Schwann cells that could be isolated from the nerve.

3. Please provide additional validation of the pdgfra+ fibroblasts or edit the text to note the need for further validation of this cluster.

Reviewer #2:

The authors have done a commendable job of addressing most of the original reviewer requests. The manuscript is certainly improved and many of the incorrect or misleading statements have been addressed as requested. The figures, labels, and legends are significantly improved in clarity.

* The requested RNA Velocity analysis is not included in the revision. The authors claim to have performed it but it did not match expectations (ie "The resulting projections do not generate evident trajectories along the dataset"). This reviewer would have preferred to examine how RNA Velocity field estimates match the existing slingshot trajectories or at least compare the presented trajectories to those created from another tool to provide confirmatory support. The expectation that the RNAVelocity findings might disagree with the presented trajectories was the motivation behind requesting this analysis. It's lack of inclusion in the final manuscript somewhat reinforces my concern regarding the accuracy/validity of the original pseudotime trajectories.

* The authors have indeed couched their statements regarding the Pdgfra-expressing fibroblast cells relative to their original statements. I am aware that their arguments are in a 'Results and Discussion' section, however, I am still less than comfortable with their speculation w/o any validation of the Pdgfra+ fibroblast population. The details of why the doublet detection algorithm (and their source of confidence in these results) is less than compelling, and the suggestion that 'damaged and/or dying cells is a contributing factor to the formation of this cluster' is supported only by circumstantial evidence of 'low quality' cells (mito read ratio and total count can by proxies for low quality cell capture, but are not generally considered proxies for cell death), not necessarily 'dying' cells. I feel that this speculation is not warranted in a Resource paper and should either be experimentally validated or removed. The manuscript would remain just as strong by merely pointing out the existence of this cluster and suggesting that it's origins require follow up analyses. But the speculation is simply not warranted w/o further supporting validation.

Reviewer #3:

In this paper, Gerber and colleagues conduct bulk and single cell RNA-Seq of mouse sciatic nerves and sorted Schwann cells to generate a developmental transcriptomic profile of Schwann cells. Overall, it is a very nicely done study and will likely serve as a good resource for the scientific community. Strengths of their approach include i) detailed developmental time course and use of 4 biological replicates for each time point, ii) combination approach of bulk sequencing from intact sciatic nerves and comparison to FAC sorted Schwann cells and addition of sc-RNA-Seq of sorted Schwann cells through the postnatal time points. The only weakness is related to the technical difficulty of obtaining higher number of Schwann cells for the scRNA-Seq component. In an intact nerve, Schwann cells account for over 70% all cells, but in the single cell sorted RNA-Seq, only 3.3% of cells were Schwann cells. This low number likely biased the results of gene expression and interpretation of the results because a "hardy" sub-population may survive the cell isolation process and not be a true representative of the whole Schwann cell population.

Authors have addressed all of my concerns. No further comments.

---

## [Author Response]

Essential revisions:1. The authors describe a publicly available searchable web interface (www.snat.ethz.ch). However, access to this resource is currently password-protected. This website needs to be made available to the reviewers in order to evaluate the utility and stability of this public resource.

The username and password were included in the original Cover Letter and in the originally Submitted Manuscript (line 777 in the originally submitted pdf file). We have now removed this information from the text of the manuscript since we plan to eliminate the password requirement upon publication of the manuscript.

2. Many of the axes are unlabeled (e.g. Figures 3 -6…). Please add labels to the figures directly.

We apologize for these omissions. The labels have now been added.

3. The clusters presented in Figure 3 Panel G (P60) are learned on a reduced dimensional embedding that the authors understand to be biased with respect to batch. Indeed, this is a particular focus of Figure 3—figure supplement 1 panel D. It is commendable that the authors note this and correct with batch correction in the supplement, however the clusters used to annotate mSC2 and mSC3 are derived from the original, biased embedding. These clusters disproportionately contain cells from each batch and therefore the differential genes between mSC2 and mSC3 in Figure 3—figure supplement 1 panel A (P60) primarily represent genes that differ between batches. This should be addressed prior to acceptance.Given the concerns about batch:cluster confounding above, the designation of mSC2 and mSC3 in the joint embedding remains concerning as well. Potentially confounding the interpretation of pseudotime in Figure 4 and Figure 4—figure supplement 1. Many of the genes identified along pseudotime for mSC3 overlap with the mSC3-specific genes from Figure 3—figure supplement 1 panel A (P60).

We show now the batch-corrected dataset at P60 in the main Figure (Figure 3, panels G, H), and in the revised Figure 3—figure supplement 1 (panels A, B). The P60 dataset without batch correction has been included in the revised Figure 3—figure supplement 1C.

We thank the reviewers for the remarks regarding the mSC clusters (see also answer to point 4). Our manuscript includes a common section for Results and Discussion. In this context, some of the discussion remarks were perceived by the reviewers rather as conclusions during the consideration of our manuscript (this notion relates similarly to point 6, see answers below). To avoid such interpretations, we have made suitable ameliorations to exert caution, also regarding informatic clusters representing myelinating Schwann cells. We now refer to these clusters commonly as myelinating Schwann cells (mSC). Accordingly, we have adjusted the cluster labels shown on tSNE projections of Figure 4, and we have included new figures that depict the original output of the clustering analysis (Figure 3—figure supplement 2 and Figure 4—figure supplement 1) to complete the information. In this revised setting, potential batch influences on the delineation of mSC1, mSC2 or mSC3 are no longer relevant (and these designations have been removed from the manuscript), since all these cluster are considered as mSC.

4. Figure 4: All axes are unlabelled, including the 'pseudotime values' along X and the gene expression levels along Y for Panel D. It is unclear that the slingshot pseudotime trajectories are resolving the best-fit paths through the data instead of potentially being biased by arbitrary cluster assignments. For example, the designation of the tSC cluster as a discrete and/or transient state is not clear from the embedding. However, the designation of this as a cluster will force slingshot to produce a trajectory through the tSC cluster when this may not be borne out directly from the data. If the authors wish to support these trajectories as appropriate paths towards specification, it is highly recommended that they further perform an RNAVelocity analysis on the joint, embedded single cell analysis in Figure 4. This will provide an 'annotation-agnostic' representation of the change in predicted state over the chosen embedding. The hope is that the velocity field mirrors the paths identified by slingshot. This will provide compelling evidence that the trajectories presented are correct and not 'winding' unnecessarily.There are some important details missing from the slingshot methods and analysis. The authors use a GAM model to identify genes with significant differential expression WRT pseudotime, but the details of the model are not presented, and the choice of significance thresholds is not described. Further, the authors do not present a list of genes identified by the GAM model, instead relying on Z-scores for previous cluster assignment specificity, (Figure 4—figure supplement 1 panel B) instead of the significance values from the pseudotime fitting. This does not use the pseudotime analysis appropriately but rather still relies on the 'aggregate' expression of genes across clusters.In general, the application of pseudotime is not complete, and the authors do not use this analysis approach to appropriately resolve trajectories and gene expression differences along these trajectories. The analysis presented is still restricted to 'cluster-level' aggregate differences.The authors interpret the "appearance of multiple mSC clusters as a peculiar feature in the tSNE plots". This is true and due, in part, to both the nature of tSNE (which does not by default preserve global structure in embeddings) and the previously noted concerns about batch (in P60) that are not addressed in the joint embedding. The authors however, choose to interpret the different mSC clusters as biologically motivated. (lines 361-368). This statement should be re-evaluated and our couched.A semantic point, the authors claim three 'biological' replicates for the 10x phase of their study per time point. Since each replicate consists of two nerves (one from male and one from female) these are not technically biological replicates but rather technical replicates.

We apologize for the omission of the labels. They have been added as requested. We have also performed RNA velocity analysis as requested. The resulting projections do not generate evident trajectories along the dataset. One potential explanation for these findings is that RNA velocity predicts the future state of cells on a timescale of hours to days, based on the dynamics of mRNAs, and the time points in the dataset in Figure 4 might be too long apart to allow a robust analysis with this bioinformatic tool. In any case, our understanding of the tSC cluster was steered by previous knowledge of Schwann cell differentiation (Figure 1A). Chronologically, iSC progressively sort large-caliber axons from bundles and become pmSC. The ensuing activation of the myelination program leads to the formation of mSC along the myelinating Schwann cell fate. Small caliber axons do not become myelinated. Instead, they are engaged by Schwann cells of the non-myelinating lineage. Using morphological knowledge as a reference, non-myelinating Schwann cells at P60 (designated nm(R)SC in the revised version of this manuscript) encircle each small caliber axon within a Remak bundle with cytoplasm. At P14, this process is ongoing but not yet complete. Based on this knowledge, we tentatively labelled the non-myelinating Schwann cells at P14 as tSC. Considering the expression patterns of established marker genes of the nm(R)SC lineage (*Ncam1*, *N1cam* and *Ngfr)*, we interpret the tSC to be cells along the non-myelinating fate at P14, which temporally precede the cells of the same fate at P60 (see also revised Figure 4C). We do not intend to state that this cluster represents a novel cell type, but it may rather represent a temporal snapshot along the non-myelinating differentiation trajectory. Therefore, the pseudotime plot depicts the clusters enriched based on prior knowledge. We have ameliorated the manuscript to reflect these issues better. The text changes include also adjustments to acknowledge the remark by the reviewers that our application of pseudotime is not exhaustive. The intention with a supervised application of pseudotime was not to resolve potentially novel trajectories. We use it to help readers visualizing the transcript abundance dynamics along the expected myelinating and non-myelinating Schwann cell lineages, crossing the center of each cluster (which show a progressive enrichment in cells of increasing mouse ages) along each respective trajectory.

The statements regarding mSC clusters have been couched as requested by the reviewers (see also answer to point 3).

We have removed the terminology of biological or technical replicates. The origin of each of the sequenced independent sample is explained in the text instead.

5. Figure 5: It is concerning that for the 10x study, SCs account for only 3.3% of the identified cell types, and yet in the bulk analysis presented in the beginning, only the SC marker genes and how they change across time points are presented. Are the SC gene signatures truly dominating in the bulk samples presented, and as suggested by the authors for the motivation for this study? This is further evidenced by the diagram provided in Figure 5—figure supplement which suggests that the vast majority of cells in this dataset should be Schwann cells. It would be very useful to know, or at least retrospectively estimate, what proportion of the gene expression signatures in the bulk study is derived from SC cells or cells of other cell types. Learning this information from the single cell data and applying this type of analysis to their bulk data would further strengthen the application and utility of this resource and demonstrate a clear need to re-evaluate the use of bulk studies in the PNS.

The proportion of Schwann cells among all cells populating the adult sciatic nerve endoneurium has been reported to be 72% of all cells (Stierli et al. 2018), in line with our own unpublished observations. The low percentage of Schwann cells in the 10x study at P60 is likely mainly due to a low extraction efficiency of viable single Schwann cells from the enzymatic and mechanic dissociation of sciatic nerves and cannot be compared to their abundance in intact sciatic nerves (this notion is now included in the revised manuscript). In bulk sequencing of sciatic nerves, however, we mechanically removed the epineurium and perineurium layers (from E17.5 onwards) and sequenced the enriched endoneurium. In this experimental setting, RNA was extracted from the tissue directly and thus, the dominant cell type that contributed to the sequenced RNA are Schwann cells. Nonetheless, other cell types in the endoneurium also contribute to the RNA levels detected in the tissue, and on its own, this experimental set cannot resolve which cell types express a given gene.

6. Figure 6: The authors 'identify clusters of proliferating fibroblasts labelled by Pdgfra expression' and reason that these clusters represent biologically interesting/unique clusters of endoneurial fibroblast-lik cells, perineuial cells, or epineurial cells, and represent a 'mysterious cluster' It is highly probable that the expression of Pdgfra in these clusters is not biological, but rather a consequence of either A) incomplete dissociation from true Pdgfra-expressing cells, or B) background signal detected in otherwise low-quality cells, (both appropriately indicated by the authors). The former is supported by the higher-than average number of genes detected in these populations (consistent with potential doublets), despite the authors use of a 'doublet detection' algorithm. The latter is supported by the higher mitochondrial read ratios observed for these clusters as well. Without supporting in situ hybridizations to support the identification and co-expression of Pdgfra in these populations, this finding should be removed or addressed in significantly greater detail. The authors claim(s) that "we cannot exclude that some cells in this cluster might also be in a hypothetical transitory stage of neural crest-derived cells, potentially expressing markers of multiple lineages on their route to differentiation" is a purely artificial explanation that is not supported by the evidence provided. Suggestion of this as an alternative explanation does not satisfy an Occom's razor explanation and must be substantiated with further experimentation.The authors state that " Clusters representing the different cell types were overall in much closer proximity to each other on the tSNE plots at P1 and P60" and use this to suggest that this may represent "potential cell lineage transitions or cell plasticity at early stages…". This is not an appropriate or valid conclusion to be drawn from tSNE plots. The meaningful proximity of clusters in a tSNE embedding cannot be directly inferred and is a common mis-interpretation of the method used. Please see discussion here for cautions and guidelines for interpretation of tSNE embeddings (https://mlexplained.com/2018/09/14/paper-dissected-visualizing-data-using-t-sne-explained).

As requested by the reviewer, we have ameliorated the text about Pdgfra-expressing cell clusters and have removed speculative discussion statements about the biology.

We agree that our original statement of “Clusters representing the different cell types were overall in much closer proximity to each other on the tSNE plots at P1 than at P60” could be interpreted as that we were inferring that the proximity between isolated clusters in tSNE projections has a meaning (which we were aware of that it does not; we were referring to clusters containing cells in direct contact with neighboring clusters). Since our statement could be misconceived, however, we have removed it from the text.

7. Given the potential utility of this dataset as a resource for the PNS community, it is strongly recommended that the authors make their analysis (scripts/code/etc) publicly available for review.

The code is available on https://github.com/suterlab/SNAT-code-used-for-primary-data-analysis. We provide also the Seurat object for each dataset, available on https://snat.ethz.ch/seurat-objects.html (Username / Password: As above). Note that with this information in place, we have removed the heatmaps previously shown in Figure 3—figure supplement 1A and Figure 4—figure supplement 1B since they can now be generated conveniently from the Seurat objects by interested researchers. Links to these resources and to the primary data in GEO have now been included in the website that accompanies this manuscript. There is also a link to the final version of the manuscript that will be activated upon publication.

[Editors' note: further revisions were suggested prior to acceptance, as described below.]

Essential revisions:1. Please address concerns about the validity of conclusions related to the pseudotime analysis raised by Reviewer #2. If additional analysis is possible, please include, otherwise modify the text to temper conclusions that can be made using this approach.

We have added the following two statements to the manuscript to indicate the limitations of our analysis and to temper the conclusions:

Related to the pseudotime analysis: “Note that the analysis was not aimed at identifying novel cell differentiation states nor trajectories, but is limited to the visualization of results obtained by a supervised bioinformatic approach in conjunction with current knowledge in the research field”.

Related to a particular point: “Note that the designation “tSC” is not intended to present SCs at P14 as a discrete cell type. In our interpretation, tSC refers rather to Remak SCs which are likely undergoing maturation”.

2. Please add additional discussion of potential bias that may arise from subsampling a subset of Schwann cells that could be isolated from the nerve.

We have added the following statement to the manuscript to discuss this issue: **“**We note that the approach of using dissociation of cells from nerves followed by FACS has the limitation of some potential subsampling of a subset of cells (Figure 2). In particular, SCs isolated from extensively myelinated nerves might not reveal the full diversity spectrum. The observed low relative extraction efficiency in this context may reflect different sensitivities of distinct SC populations to the experimental procedures, associated with the potential of subsampling”.

3. Please provide additional validation of the pdgfra+ fibroblasts or edit the text to note the need for further validation of this cluster.

We have followed the advices and have replaced the section in question accordingly with the statement: “Determination of the origins and validation of this cluster requires further investigations”.

Reviewer #2:The authors have done a commendable job of addressing most of the original reviewer requests. The manuscript is certainly improved and many of the incorrect or misleading statements have been addressed as requested. The figures, labels, and legends are significantly improved in clarity.* The requested RNA Velocity analysis is not included in the revision. The authors claim to have performed it but it did not match expectations (ie "The resulting projections do not generate evident trajectories along the dataset"). This reviewer would have preferred to examine how RNA Velocity field estimates match the existing slingshot trajectories or at least compare the presented trajectories to those created from another tool to provide confirmatory support. The expectation that the RNAVelocity findings might disagree with the presented trajectories was the motivation behind requesting this analysis. It's lack of inclusion in the final manuscript somewhat reinforces my concern regarding the accuracy/validity of the original pseudotime trajectories.

We had uploaded the code for the RNA velocity analysis on which our statement was based on *Github* and we have now complemented this information further, including output data. See (https://github.com/suterlab/SNAT-code-used-for-primary-data-analysis).

* The authors have indeed couched their statements regarding the Pdgfra-expressing fibroblast cells relative to their original statements. I am aware that their arguments are in a 'Results and Discussion' section, however, I am still less than comfortable with their speculation w/o any validation of the Pdgfra+ fibroblast population. The details of why the doublet detection algorithm (and their source of confidence in these results) is less than compelling, and the suggestion that 'damaged and/or dying cells is a contributing factor to the formation of this cluster' is supported only by circumstantial evidence of 'low quality' cells (mito read ratio and total count can by proxies for low quality cell capture, but are not generally considered proxies for cell death), not necessarily 'dying' cells. I feel that this speculation is not warranted in a Resource paper and should either be experimentally validated or removed. The manuscript would remain just as strong by merely pointing out the existence of this cluster and suggesting that it's origins require follow up analyses. But the speculation is simply not warranted w/o further supporting validation.

See point 3 of Essential revisions.